# Impacts of ocean warming on fish size reductions on the world's hottest coral reefs

Jacob L. Johansen [1,2] ✉, Matthew D. Mitchell[2], Grace O. Vaughan[2,3], Daniel M. Ripley [2,4], Holly A. Shiels [4] & John A. Burt [2,5]

The impact of ocean warming on fish and fisheries is vigorously debated. Leading theories project limited adaptive capacity of tropical fishes and 14-39% size reductions by 2050 due to mass-scaling limitations of oxygen supply in larger individuals. Using the world's hottest coral reefs in the Persian/Arabian Gulf as a natural laboratory for ocean warming - where species have survived >35.0 °C summer temperatures for over 6000 years and are 14-40% smaller at maximum size compared to cooler locations - we identified two adaptive pathways that enhance survival at elevated temperatures across 10 metabolic and swimming performance metrics. Comparing *Lutjanus ehrenbergii* and *Scolopsis ghanam* from reefs both inside and outside the Persian/Arabian Gulf across temperatures of 27.0 °C, 31.5 °C and 35.5 °C, we reveal that these species show a lower-than-expected rise in basal metabolic demands and a right-shifted thermal window, which aids in maintaining oxygen supply and aerobic performance to 35.5 °C. Importantly, our findings challenge traditional oxygen-limitation theories, suggesting a mismatch in energy acquisition and demand as the primary driver of size reductions. Our data support a modified resource-acquisition theory to explain how ocean warming leads to species-specific size reductions and why smaller individuals are evolutionarily favored under elevated temperatures.

As oceans warm, our ability to sustainably manage and protect important species depends on how well we can predict temperature-driven processes operating at individual, population, and community levels[1,2]. Many aquatic ectotherms, especially those adapted for the Arctic and tropics, are expected to be highly sensitive to temperature increases beyond those in which they have evolved[3–5]. Given the critical importance of aquatic resources for human survival, the likely consequences of rising temperatures for species fitness and productivity are vigorously debated[6,7].

The Temperature-Size rule postulates that ectotherms living in warmer conditions grow faster as juveniles but attain smaller adult body sizes[8,9]. This phenomenon of reducing body size with increasing temperature has been observed from the poles to the equator in organisms ranging from bacteria to vertebrates[8,10,11], and is particularly pronounced in aquatic ectotherms[12–16]. In accordance with the Temperature-Size rule, the maximum size of many fishes has already declined by 5–29% over the last few decades as a direct consequence of ocean warming[17–19], and global fish populations are projected to decline in mass by a further 14–24% by 2050[20,21]. The largest such declines are expected in warm, tropical habitats, where coral reef fish may decline in body mass by as much as ~39% by 2050[20]. This 'shrinking of fishes' phenomenon is expected to have profound implications for fisheries productivity worldwide[22–24], as well as intra-[25] and inter-specific interactions[24,26], community structure, population

[1]Hawaii Institute of Marine Biology, University of Hawaii at Manoa, Honolulu, HI, USA. [2]Marine Biology Laboratory, New York University Abu Dhabi, Abu Dhabi, United Arab Emirates. [3]BiOrbic, Bioeconomy SFI Research Centre, O'Brien Centre for Science, University College Dublin, Dublin, Ireland. [4]Division of Cardiovascular Sciences, Faculty of Biology, Medicine and Health, The University of Manchester, Manchester, United Kingdom. [5]Mubadala ACCESS Center, New York University Abu Dhabi, Abu Dhabi, United Arab Emirates. ✉e-mail: jacoblj@hawaii.edu

demography, and ecosystem trophic structure and functioning[3,27–29]. Yet, despite the ubiquity and importance of the Temperature-Size rule phenomenon, the causal mechanisms underpinning these changes remain unclear and hotly contested[1,2,30–34].

Two competing "oxygen-limitation" hypotheses have gained traction to explain the Temperature-Size rule for aquatic ectotherms, both centered on the idea that oxygen supply and demand do not scale equally with body size[35]. The Gill Oxygen Limitation hypothesis (GOL[36]) argues that body size in fish is limited by gill area; gills are seen as a two-dimensional surface with limited growth potential due to geometrical constraints, and oxygen supply therefore cannot keep up with three-dimensional increases in body size. In this theory, larger-bodied individuals will be unable to compensate for increased metabolic demands associated with high temperature due to the gills providing decreasing amounts of oxygen per unit body mass. As a result, fishes will reach maximum size when their gills can only meet the oxygen demands of basal metabolism, i.e. as aerobic surplus becomes inadequate to support further growth[21,31,37].

The second prevailing body-size hypothesis is called "Maintain Aerobic Scope and Regulate Oxygen Supply" (MASROS[2,38]). In this hypothesis, evolution is assumed to have modified growth trajectories to avoid the loss of aerobic surplus predicted by GOL[2]. Body size is instead optimized to maintain a safety margin of oxygen supply beyond basal demand that is adequate to fuel critical ecological activities needed for survival. Here, developmental plasticity is thought to ensure that body size is optimized for a given environmental temperature so that animals can maintain critical performance metrics when reaching maximum size (i.e. animals stop growing when additional growth would compromise critical performance metrics[39]). The MASROS theory is exemplified by the 'Ghost of Oxygen-limitation Past' hypothesis, which considers maximal body size to have evolved in response to the temperature and oxygen limitation experienced by ancestors, favoring genotypes that grow to a smaller size in warm water[2]. Despite significant global attention given to both hypotheses, there is currently no consensus as to the capacity of either hypothesis to provide a universal explanation for the Temperature-Size rule[21,24,31,34,35,37,39,40].

In fishes, critical performance metrics include digestion, growth and reproduction[34,39,41,42], as well as swimming, which is usually the most energetically costly function that fish perform[43], but is required for migration, foraging and predator evasion[44,45]. The aerobic surplus fueling these functions is driven by two extremes: the minimum rate of oxygen supply a fish needs to maintain homeostasis, termed the standard metabolic rate (SMR[46]), and the highest achievable rate of oxygen supply, usually obtained during exhaustive exercise, termed the maximum metabolic rate (MMR[47]). The difference between SMR and MMR (i.e. MMR−SMR) then defines the available aerobic surplus, termed aerobic scope[40].

The aerobic scope of fish and their surplus capacity for performance may be impacted by both body mass and temperature. At optimal temperature for a species (i.e. the temperature at which a species can perform maximally), SMR, MMR and aerobic scope usually increase allometrically with body mass by exponents 0.8–0.9[48–50]. As a result, the magnitude by which MMR stays above SMR (defined as the factorial aerobic scope) often remains fixed in relation to body mass[31]. As temperatures rise above optimum, however, these patterns change and become less clear. The mass scaling exponents of SMR have been shown to be species-specific across a number of fish species and can either be stable or decrease across temperatures[51–53]. By comparison, the effect of mass scaling and temperature changes on MMR and aerobic scope in fishes has only been examined in a few temperate species to date[52–54], but preliminary evidence from these studies suggests that MMR scales with a declining exponent at higher temperatures, causing aerobic scope to peak at lower temperatures for large individuals. Likewise, little is known for scaling of swimming

performance, but given the necessity for aerobic surplus, locomotor performance is likely to follow metabolic responses across temperatures as previously shown for e.g., coral reef fishes[55–58].

There is little doubt that species are already responding to changing environmental conditions including rising temperatures[59], and that the maximal body size of many species appears to be reducing[17–20,22,23,60]. However, validation of the proposed theoretical explanations is complicated by the need to evaluate patterns over multigenerational timescales, and the fact that fishes may simultaneously optimize performance efficiency through exposure to elevated temperatures. For instance, many fishes are expected to have high thermal sensitivity to rising ocean temperatures, causing SMR to increase near exponentially (i.e. with a $Q_{10}$ of ~2–3)[48]. Likewise, temperature has a significant impact on fish muscle function, altering maximal contraction frequency and power, which in turn affects the aerobic efficiency of swimming[61–64]. Understanding how aerobic demands may respond to elevated temperatures though time is therefore paramount for species response projections and testing of the GOL and MASROS hypotheses. There is an urgent need for dedicated multigenerational experimental studies that are able to accurately quantify aerobic energy budgets and critical performance metrics, such as swimming, across known temperature conditions[65,66]. Importantly, the use of populations that have naturally evolved in temperatures expected globally in the next century are most likely to yield results that are applicable for climate change predictions and theories[2,31,37,50,67].

Here, we utilize the hottest coral reefs on Earth as a window to the future. Coral reefs in the Southern Persian/Arabian Gulf today experience typical summer water temperatures reaching 36.0 °C[68–71], conditions which are comparable to worst-case business-as-usual ocean warming projections for many tropical coral reefs globally by 2100[6]. Although the Persian/Arabian Gulf reefs have been exposed to these elevated temperatures for over 6000 years[68,72], fewer than 60 known species of reef fish have managed to survive there[71,73]. In contrast, less than 300 km away in the adjacent and biogeographically connected Gulf of Oman more than 500 species of fishes exist under thermal conditions typical of present-day coral reefs (≤32.0 °C summer maximum[71,73–76],). Importantly, fishes within the Persian/Arabian Gulf are also 14–40% smaller at maximum size compared to populations of the same species and age within the Gulf of Oman (including cryptic and larger bodied fishes not targeted by fishers[71,77]), highlighting that the projected 'shrinking of fishes' phenomenon[20] is already in full effect within this ecosystem.

Here, we use the Persian/Arabian Gulf as a natural laboratory for ocean warming to test the consequences of elevated temperatures on metabolism and swimming performance both within and across species. By comparing fishes surviving under the elevated temperatures within the Persian/Arabian Gulf to those from the more benign conditions within the Gulf of Oman, we aim to elucidate 1) whether reef fishes in the Persian/Arabian Gulf have managed to improve metabolic and swimming performance metrics to support survival at elevated temperatures; and 2) whether theoretical GOL and MASROS explanations for size reductions match observed patterns in the field. Based on previous studies showing high thermal sensitivity and limited acclimation and adaptive capacity in tropical reef fishes[3–5], we did not predict significant performance improvements in the Persian/Arabian Gulf fishes relative to Gulf of Oman fishes. However, based on the GOL and MASROS explanations for the Temperature-Size rule[2], we predicted that the mass scaling exponents of metabolic (GOL) and locomotor (MASROS) performance should reduce at elevated temperatures, causing larger fish to experience greater reductions in performance with each degree increase in temperature. Contrary to expectations, we found evidence of divergent adaptive pathways facilitating performance and survival to 35.5 °C, but these did not fully comply with mass-scaling predictions. Instead, our data supports a

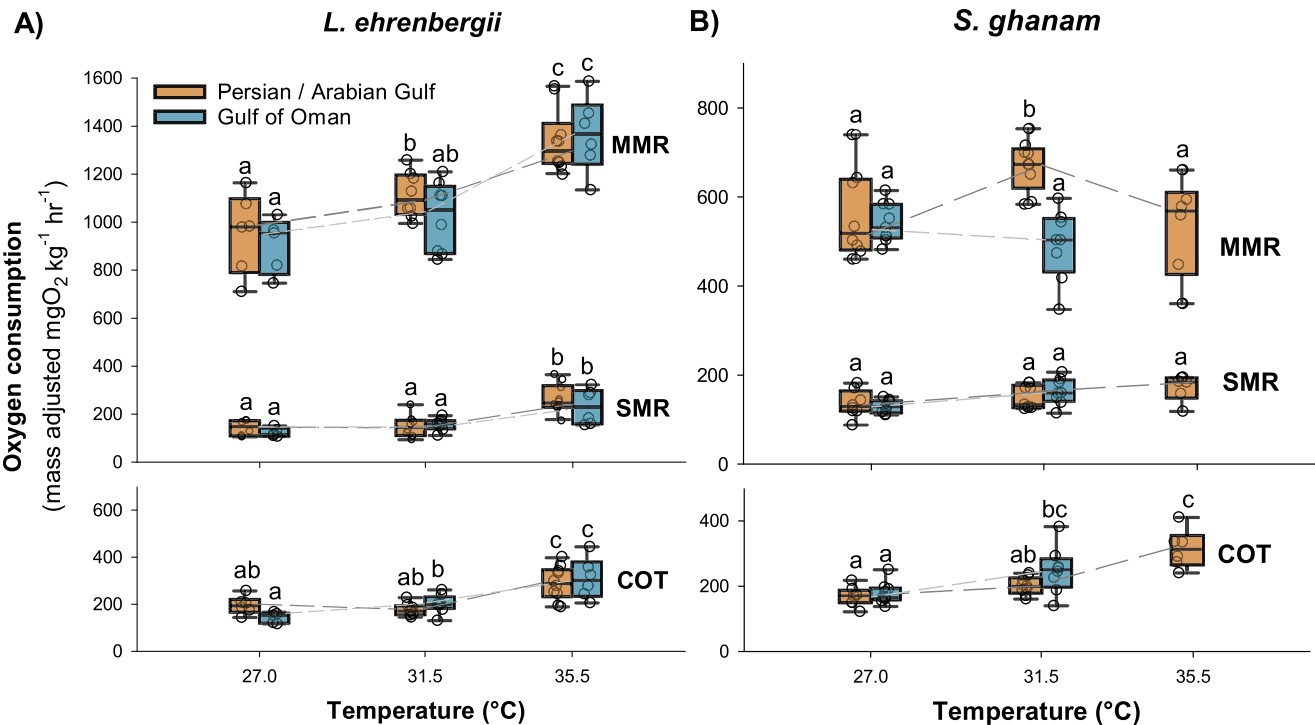

**Fig. 1 | Box plot of metabolic performance across temperatures of two species of coral reef fishes from the Persian/Arabian Gulf and Gulf of Oman.** Panels **A** and **B** show mass-adjusted maximum metabolic rate (MMR), standard metabolic rate (SMR) and cost of transport (COT) of *Lutjanus ehrenbergii* and *Scolopsis ghanam*, respectively. Note, there are no data for Gulf of Oman *S. ghanam* at 35.5 °C as no individual could be acclimated to this temperature. Each box depicts median, 25 and 75 percentiles, and dashed lines depicts cross-temperature responses. Raw data are shown as circles behind each box, with $n = 43$ and $n = 40$ for *L. ehrenbergii* and *S. ghanam*, respectively. Error bars depict 95 percentiles and significant differences within metrics are marked by lower case letters. Source Data are provided in Supplementary Data 1.

modified theory suggesting that size reductions and survival as oceans warm are inherently linked to a mismatch in energy acquisition and demand.

## Results

### Cross regional temperature performance

We did not predict significant performance improvements in the Persian/Arabian Gulf fishes relative to Gulf of Oman fishes. Accordingly, *L. ehrenbergii* showed no regional difference in performance and no temperature driven loss of performance. Specifically, there was no difference in metabolic performance (i.e. SMR, MMR, aerobic scope, Cost of Transport [COT]) between Persian/Arabian Gulf and Gulf of Oman individuals at any temperature ($F_{1,37} = 1.194$, $p = 0.282$, Adj $\eta^2 p = 0.005$, Supplementary Data 2). Instead, *L. ehrnebergii* showed consistent cross-regional increases in these metabolic metrics with rising temperature from 27.0 to 35.5 °C (Fig. 1, Supplementary Data 2), concurrent with a low temperature sensitivity quotient ($Q_{10}$) of 2.04 across 27.0 °C to 35.5 °C. This species also showed no difference in swimming performance (i.e. optimal, burst, and critical swimming speeds) between regions at any temperature ($F_{1,37} = 0.372$, $p = 0.546$, Adj $\eta^2 p = 0.000$), and no significant changes across temperatures (Fig. 2, Supplementary Data 2). Within temperature comparisons of swimming kinematics (Amplitude, Frequency, Strouhal) also showed no significant difference among regions for any temperature, and no change in kinematic efficiency (i.e. Strouhal number) across temperature (Fig. 3, Supplementary Data 2).

Contrary to predictions, *S. ghanam* showed significant regional differences in performance across temperatures, culminated by the fact that only Persian/Arabian Gulf individuals could be acclimated to the 35.5 °C treatment (Fig. 1, Supplementary Data 3). Specifically, while the metabolic performance of Gulf of Oman individuals remained stable from 27.0 to 31.5 °C, AG individuals showed significant increases

in MMR and aerobic scope leading to broad regional differences at 31.5 °C ($F_{1,35} = 6.307$, $p = 0.017$, Adj $\eta^2 p = 0.130$, Fig. 1, Supplementary Data 3). From 31.5 to 35.5 °C, Persian/Arabian Gulf individuals demonstrated a right-shifted thermal window, with MMR returning to 27.0 °C levels and metabolic performance metrics at 35.5 °C showing no overall difference from Gulf of Oman individuals at 31.5 °C (Fig. 1, Supplementary Data 3). This species also showed a lower-than-expected temperature sensitivity quotient ($Q_{10}$) of 1.69 across 27.0 °C to 31.5 for Gulf of Oman individuals, and 1.33 across 27.0 to 35.5 °C for Persian/Arabian Gulf individuals, concurrent with a significant increase in COT (Fig. 1, Supplementary Data 3).

There was also a significant difference in swimming performance between regions for this species ($F_{1,35} = 19.380$, $p < 0.001$, Adj $\eta^2 p = 0.340$). Specifically, Persian/Arabian Gulf individuals had significantly higher optimal, burst, and critical swimming speeds at 31.5 °C (Fig. 2, Supplementary Data 3). These individuals also showed no change in swimming performance from 27.0 to 31.5 °C and were able to swim consistently at 35.5 °C, again denoting a right-shifted thermal window of performance (Fig. 2). In comparison, the Gulf of Oman individuals suffered significant reductions in optimal, burst, and critical swimming speeds with rising temperature from 27.0 to 31.5 °C ranging from 16.6 to 22.4% (Fig. 2, Supplementary Data 3). For Swimming kinematics, Amplitude differed between regions at 27.0C (Supplementary Data 3) but there was no change in Frequency or swimming efficiency (Strouhal number) between regions at any temperature (Fig. 3, Supplementary Data 3).

### Effect of temperature on mass-scaling

We predicted that larger fish should experience greater reductions in metabolic performance (following the GOL hypothesis) and swimming performance (following the MASROS hypothesis) with each degree increase in temperature. However, contrary to predictions, *L.*

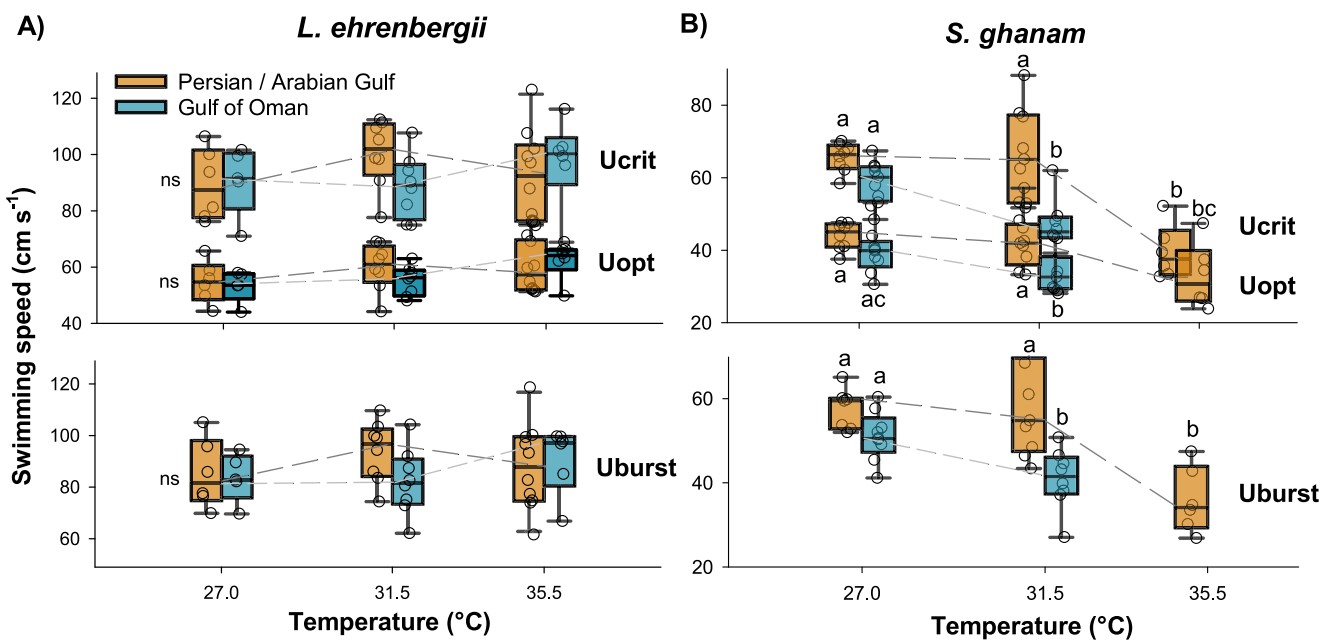

**Fig. 2 | Box plot of swimming performance across temperatures of two species of coral reef fishes from the Persian/Arabian Gulf and Gulf of Oman.** Panels **A** and **B** shows adjusted critical swimming speeds ($U_{crit}$), optimal swimming speed ($U_{opt}$) and burst swimming speed ($U_{burst}$) of *Lutjanus ehrenbergii* and *Scolopsis ghanam*, respectively. Note, there are no data for Gulf of Oman *S. ghanam* at 35.5 °C as no individual could be acclimated to this temperature. Each box depicts median, 25 and 75 percentiles, and dashed lines depicts cross-temperature responses. Raw data are shown as circles behind each box, with $n = 43$ and $n = 40$ for *L. ehrenbergii* and *S. ghanam*, respectively. Error bars depict 95 percentiles and significant differences within metrics are marked by lower case letters. Source Data are provided in Supplementary Data 1.

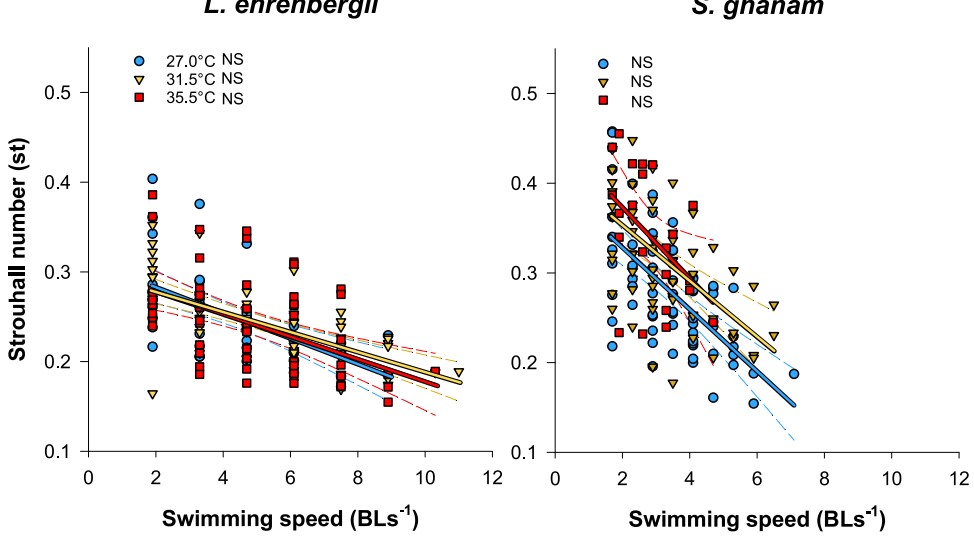

**Fig. 3 | Regression plot of swimming speed kinematics modeled for two species of coral reef fishes from the Persian/Arabian Gulf and Gulf of Oman.** Panels show regressions of Strouhall number (St) of *Lutjanus ehrenbergii* and *Scolopsis ghanam*, respectively. Blue, orange and red lines and markers represent pooled 27.0 °C, 31.5 °C and 35.5 °C data. Dashed lines depict estimated 95 confidence intervals. There were no significant (NS) differences in the regression within either species. Source Data are provided in Supplementary Data 1.

*ehrenbergii* was not limited by mass at elevated temperatures and there were no recorded negative consequences for larger individuals (Fig. 4, Supplementary Data 2). Specifically, *L. ehrenbergii* showed persistent increases in metabolic performance metrics with increasing mass at all temperatures (Figs. 4 and 5, Supplementary Data 2). This species also showed no changes to mass-scaling slopes across temperatures, except for a steeper positive increase in COT with increasing mass at 35.0 °C than at 27.0 °C and 31.5 °C (Fig. 5, Supplementary Data 2). For swimming performance, $U_{crit}$ increased weakly with

increasing mass at 27.0 °C and remained stable across mass at 31.5 and 35.5 °C, but showed no overall differences in mass-scaling slopes across temperatures and no negative effect on larger individuals (Fig. 4, Supplementary Data 2).

Contrary to GOL predictions, *S. ghanam* was also unaffected by mass in metabolic performance, but did follow MASROS predictions of being limited by mass in swimming performance. Specifically, across temperatures this species showed overall increases in metabolic performance metrics with increasing mass and no changes in mass-scaling

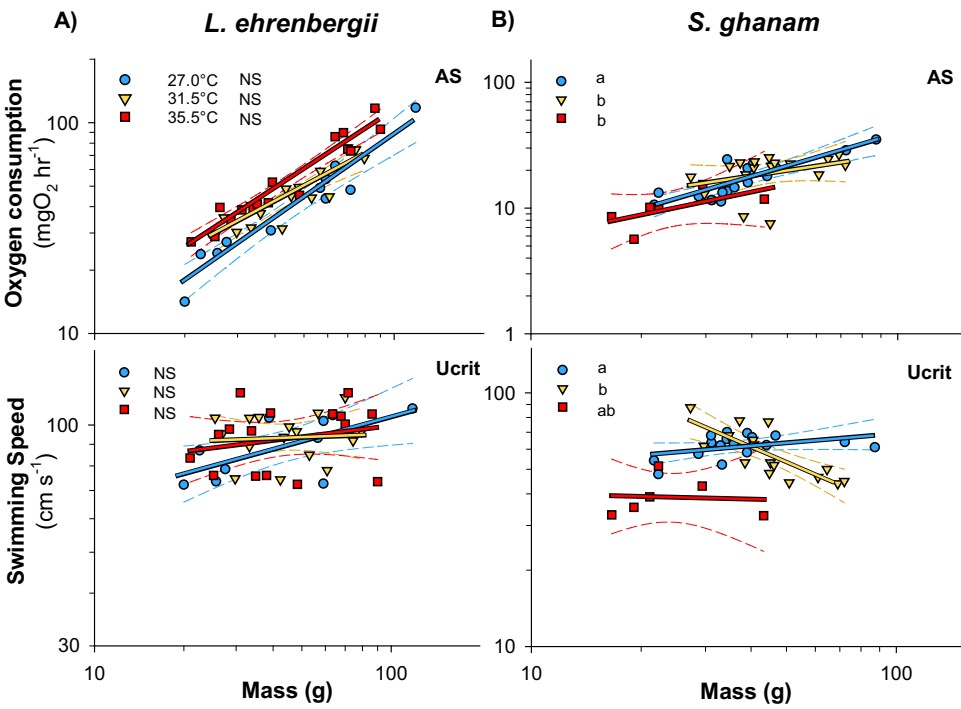

**Fig. 4 | Regression plot of mass scaling relationships modeled for two species of coral reef fishes from the Persian/Arabian Gulf and Gulf of Oman.** Panels **A** and **B** shows regressions of aerobic scope (AS) and critical swimming speed (U$_{crit}$) of *Lutjanus ehrenbergii* and *Scolopsis ghanam*, respectively. Blue, orange and red lines and markers represent pooled 27.0 °C, 31.5 °C and 35.5 °C data. Dissimilar letters denote significant differences in slope, while dashed lines depict estimated 95 confidence intervals. NS denotes non-significance. Source Data are provided in Supplementary Data 1.

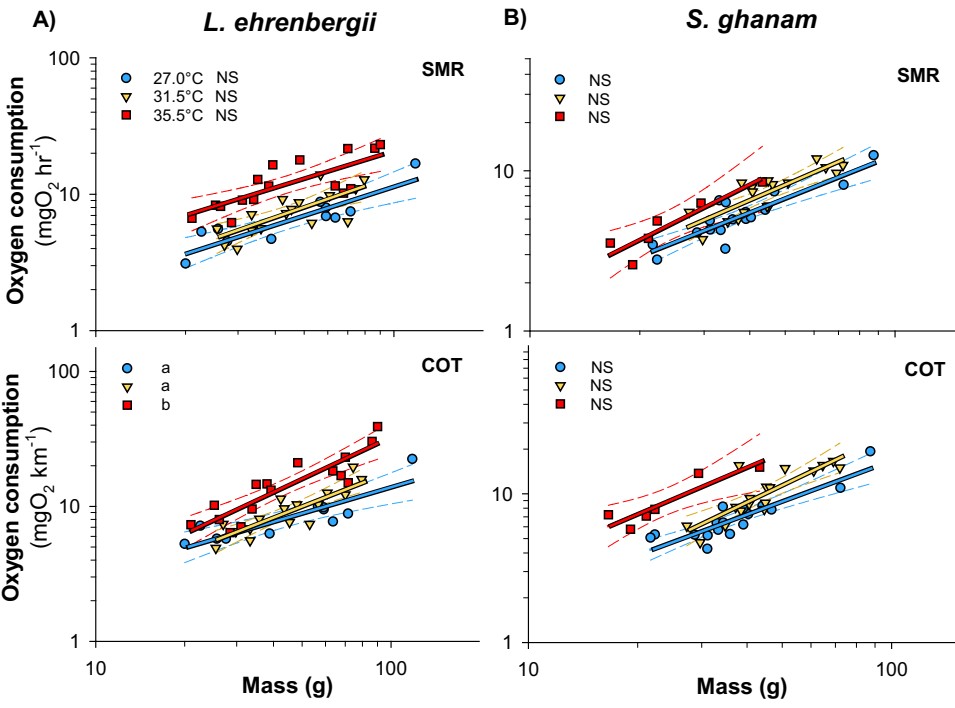

**Fig. 5 | Regression plot of mass scaling relationships modeled for two species of coral reef fishes from the Persian/Arabian Gulf and Gulf of Oman.** Panels **A** and **B** shows regressions of standard metabolic rate (SMR) and cost of transport (COT) of *Lutjanus ehrenbergii* and *Scolopsis ghanam*, respectively, as estimates of total costs of living. Blue, orange and red lines and markers represent pooled 27.0 °C, 31.5 °C and 35.5 °C data. All regressions have significant positive slopes highlighting increasing cost of living with size. Dissimilar letters denote significant differences in slope, while dashed lines depict estimated 95 confidence intervals. NS denotes non-significance. Source Data are provided in Supplementary Data 1.

slopes of SMR and COT (Fig. 5, Supplementary Data 3). Only the mass-scaling of aerobic scope showed a temperature effect, by increasing significantly more with mass at 27.0 °C than at 31.5 °C or 35.5 °C (Fig. 4, Supplementary Data 3). In accordance with predictions for MASROS, however, *S. ghanam* was limited by mass in swimming performance. This species showed no significant effect of mass on $U_{crit}$ at 27.0 °C, but a significant negative effect of mass at 31.5 °C followed by a mean reduction in $U_{crit}$ at 35.5 °C where only the smaller Persian/Arabian Gulf individuals remained (Fig. 4, Supplementary Data 3). As a consequence, the mass-scaling slope of $U_{crit}$ differed between 27.0 °C and 31.5 °C, highlighting a significant negative effect on larger individuals at elevated temperatures (Fig. 4, Supplementary Data 3).

## Discussion

The consequences of rising ocean temperatures for fish fitness and productivity is a topic of intense debate[6,7,17,20,22,59]. Prevailing hypotheses project 14–39% size reductions of tropical coral reef fishes by 2050[3–5,20] due to the combined effect of low adaptive potential and mass-scaling limitations of oxygen supply in larger individuals[2,33,78]. Utilizing coral reefs in the Persian/Arabian Gulf as a natural laboratory for elevated ocean temperatures, our data reveals two adaptive pathways across 10 metabolic and swimming performance metrics that appear to have facilitated survival, including a lower-than-expected rise in basal metabolic demands, and a right-shifted thermal window enabling individuals to maintain aerobic scope and critical swimming speeds into elevated temperatures. Importantly our findings challenge leading theoretical explanations for warming-induced size reductions in fishes, as the species and populations examined here did not fully comply with mass-scaling predictions, suggesting recorded size reductions are driven by factors other than oxygen supply limitations in larger individuals.

### Performance

Although more than 500 species of reef fish occur in the Gulf of Oman, fewer than 60 species are found in the thermally extreme Persian/Arabian Gulf. This disparity persists despite a close proximity (i.e. less than 300 km), a direct connection, and a gradual transition in thermal maxima between the two[68,79]. The fact that species have had over 6000 years to acclimate and adapt to Persian/Arabian Gulf conditions[72] but less than 12% have managed to do so, seems to confirm that many tropical fishes have low adaptive capacity and, hence, likely also have a low chance of adapting to the +3 °C warming that has been projected for many coral reefs globally by 2100[80].

A major problem anticipated for tropical coral reef fish as oceans warm is the ability to maintain physiological performance necessary for ecological survival. Even for species that can maintain aerobic surplus to fuel critical activities, previous empirical and theoretical research has suggested that rising temperatures can significantly reduce swimming performance[55,81], likely driven by the diminished function of muscles, tissues and oxygen transport systems at temperatures above optimum[82–84]. Swimming is central to all ecological activities of fish, and performance reduction in species that e.g. hunt for mobile prey is likely to reduce prey encounter and capture rates, particularly of more thermally tolerant prey[85,86]. Coupled with a predicted exponential increase in metabolic demands at higher temperatures[87–89], a likely outcome for affected species is diminished energy available for maintenance, growth, and reproduction[43,86,90]. It is clear that changes in physiological performance, regardless of cause, have the capacity to significantly alter population fitness and survival in regions where ecological functions cannot be upheld.

Despite predictions, neither species showed substantial increases in basal metabolic demands (SMR) and both showed a lower-than-expected temperature sensitivity quotient ($Q_{10}$) of 1.33–2.04 across the range of temperatures examined here. Given the ectothermic nature of most fishes and direct correlations between temperature and biochemical reaction rates[41], there is an expectation of exponentially rising metabolic demands as a consequence of ocean warming. However, it is also well known that most organisms exhibit phenotypic plasticity in the metabolic response to environmental challenges, which may allow suppressed post-acclimation basal demands across temperature within the constraints of stoichiometry in relation to body size[41,48,87,91]. As such, limited increases in SMR and energetic demand may promote survival in high temperature habitats and although direct empirical evidence is limited, suppressed SMR has been shown in laboratory zebrafish acclimated to elevated temperatures over six generations (see Wootton et al.[1]). Intriguingly, based on evaluations of thermal sensitivity in hundreds of fish species across latitudes, the average post-acclimatization $Q_{10}$ equates to ~1.65–1.91[41,48,91,92] across species and 2.40 within species[48,93]. Our results closely resemble these post-acclimatization values with a mean $Q_{10}$ of 1.71, supporting the notion that our fishes were indeed fully acclimated to the temperatures ranges examined, and that reductions in basal metabolic demands may aid survival of reef fishes as temperatures rise.

Beyond limitations in basal metabolic demands, we predicted similar performance between Persian/Arabian Gulf and Gulf of Oman fishes, meaning that only species with high intrinsic thermal tolerance should persist within the Persian/Arabian Gulf. Yet, our findings suggest that two divergent pathways have facilitated fitness and survival of fishes in the Persian/Arabian Gulf. Explicitly, *L. ehrenbergii* followed predictions by showing no population level signs of adaptation to life at the elevated temperatures found within the AG, instead appearing to thrive in the Persian/Arabian Gulf due to a high intrinsic species-specific tolerance to temperatures reaching 35.5 °C. Indeed, concurrent with increasing metabolic demand for activity (i.e. cost of swimming) caused by rising temperatures[87–89], this species showed an analogous increase in surplus energy available to fuel ecological functions both in Persian/Arabian Gulf individuals collected at 35.5 °C and in and Gulf of Oman individuals after short-term acclimation to 35.5 °C. Specifically, as COT and cost of critical swimming (metabolic rate at maximum critical swimming speed) increased across temperatures, so too did the aerobic scope. As a result, there was no evidence of oxygen limitation in this species and no change in any swimming performance metric examined (neither optimal, burst, or critical swimming speeds or kinematics), highlighting that the swimming dependent activities needed by this species for survival at +3 °C (including foraging and predator evasion) were well within its existing performance window.

Contrary to *L. ehrenbergii*, *S. ghanam* showed clear signs of a population level response to survival at the elevated temperatures of the Persian/Arabian Gulf both in terms of oxygen supply capacity and swimming performance. While this species also showed increasing metabolic demand for swimming with rising temperature, only the Persian/Arabian Gulf individuals were able to maintain aerobic scope up to 35.5 °C. This metabolic surplus allowed the Persian/Arabian Gulf individuals to fuel critical swimming activities at 35.5 °C, whereas the Gulf of Oman individuals demonstrated a complete inability to be acclimated to the same temperature. Interestingly, the Gulf of Oman individuals also suffered >20.0% reductions in aerobic scope and concurrent 17–22% reductions in all swimming performance metrics at 31.5 °C, suggesting that the ecology of this species allows individuals to survive prolonged periods of reduced swimming performance during summer. Indeed, previous work has shown that some fishes in the Persian/Arabian Gulf region broaden their diet in the spring but then narrow their diets in the summer[94], while others upregulate foraging and energy storage in the spring, but then downregulate costly swimming activities during summer[95]. This latter trait was mirrored in the Persian/Arabian Gulf *S. ghanam*, as there were no differences in metabolic and swimming performance of the Gulf of Oman individuals at 31.5 °C and the Persian/Arabian Gulf individuals at 35.5 °C, suggesting that population level responses to elevated temperatures have led

to a significant right-shift in the thermal performance window of this species[90], but no other changes in the ecological performance or survival strategy during peak summer. As a result, it appears that some reef fish, even if a minority among tropical fishes, can adjust to survive at temperatures substantially above current maxima for most coral reefs globally.

## Size-theory

A major consequence of survival at elevated temperature is the universal reduction in maximum body size recorded for species spanning small cryptobenthic fishes to larger reef piscivores in the Persian/Arabian Gulf[71,77]. Owing to the implication of rising ocean temperature on ecosystem productivity and fisheries yields worldwide, a method for predicting outcomes for individual species would be very useful. Although several leading hypotheses have attempted to explain this phenomenon, empirical support has been lacking. To our knowledge, this study provides the first comprehensive examination of wild fish species which have survived for several thousands of years under temperatures similar to those projected for coral reefs globally by 2100. In accordance with established hypotheses, we predicted that the mass scaling exponents of metabolic and locomotor performance should reduce at elevated temperatures, causing larger fish to experience greater reductions in performance with each degree increase in temperature than smaller fish. Yet, our results did not conform to expectations and did not follow hypothetical predictions for body size reductions due to oxygen supply limitations[2,35,36,38].

Oxygen supply has long been considered a key element for cellular to organismal growth, and previous evidence has suggested that the capacity for growth can be limited by oxygen in aquatic ectotherms[21]. For instance, gill surface area in some fish species decreases relative to body size as individuals grow, theoretically rendering larger individuals less able to support increasing temperature-driven metabolic demands[20,36,53,54]. Gill oxygen-limitation has therefore been proposed as a potential explanation for the size reductions (GOL[36]), whereby decreased oxygen solubility and increasing metabolic demands at warmer temperatures could cause oxygen supply to a larger body to become increasingly difficult. Indeed, if oxygen supply to large body size is compromised at higher temperatures, smaller body size would be adaptive[37]. However, several studies have challenged the claim that oxygen supply could limit growth and body size in gill breathing ectotherms, based on the notion that gill oxygen uptake can be modulated via a number of pathways including increased lamellar recruitment[31,78]. Similarly, Audzijonyte et al.[37], highlighted that existing evidence for GOL largely comes from short duration studies which could lead to erroneous conclusions as it may take numerous generations before adaptive metabolic responses emerge[1,93]. Our findings support these latter notions as we did not find evidence of limitations in oxygen supply with mass in Persian/Arabian Gulf fishes. Conversely, we found either an increase or no change in aerobic scope with size in the species and populations examined, suggesting that maximal oxygen uptake is more likely to reflect the evolved maximal demand needed by a given species for ecological survival rather than a hard physiological limit on oxygen supply capacity as suggested by GOL[1,31,37,49,96]. Similarly, recent studies of growth rates in fishes found no effect of maximum water temperature when oxygen demand is at its highest[50,77,97], further supporting the notion that oxygen supply capacity is not a limiting factor for growth[40,98]. Consequently, while oxygen supply undoubtedly plays a pivotal role for survival, the GOL does not appear to provide a convincing universal explanation for the "shrinking of fishes" phenomenon (see similar conclusions by Scheuffele et al.[33] and Wootton et al.[1]).

The MASROS theory provides a more nuanced approach by suggesting that developmental plasticity ensures that body size is optimized for a given environmental temperature to maintain adequate AS

(i.e. aerobic safety margin) to fuel critical performance metrics, such as swimming and digestion[2,38]. Thus, body size should reduce at elevated temperatures to avoid impaired performance of larger individuals[2,37,38]. Theoretical modeling has provided some support for this[35], but empirical support has been lacking due to the rarity of datasets that include multigenerational acclimation to several temperatures, naturally acclimated and adapted populations, and a sufficiently large range of body masses for a single species to determine the scaling exponents for performance traits[67,99,100]. We expected data from our two species across two regions to provide collective support of MASROS in the form of reduced $U_{crit}$ across temperature and mass. Instead, we found that regardless of temperature, *L. ehrenbergii* maintained its $U_{crit}$ across mass, despite the fact that this species has shrunk by ~19% in the Persian/Arabian Gulf[77]. Only *S. ghanam* showed limited support for MASROS. While this species maintained its $U_{crit}$ across mass at 27.0 °C, both Persian/Arabian Gulf and Gulf of Oman individuals demonstrate a mass-specific reduction in $U_{crit}$ at 31.5 °C. Interestingly, this pattern disappeared at 35.5 °C where only Persian/Arabian Gulf individuals remained, although the fact that only the Persian/Arabian Gulf individuals could be collected and directly tested at 35.5 °C renders some uncertainty for the mass scaling exponent at this temperature. Overall, these results provide limited support for MASROS in one species but refutes MASROS as a universal explanation for size-reduction in fishes. Importantly, these results also highlight the importance of multi-species studies, as a study focused solely on *S. ghanam* would have missed the fact that AG fishes have reduced in maximum size irrespective of the ability to maintain aerobic scope and swimming performance at larger sizes, and that other factors must be implicated in body size reductions at elevated temperatures.

## Resource limitation

It is possible that body size reductions may represent a life-history trade-off, whereby reduced metabolic demands enhance survival and result in a competitive advantage for smaller individuals with less absolute resource requirements. Several studies have suggested temperature- and size-dependent resource limitation as a causative factor for declining body sizes in nature[101–106]. Optimal body size should match resource demand to resource supply on a *per-capita* basis[107]. If temperature affects the *per-capita* resource demand and supply at size-dependent rates, then negative temperature-size relationships could arise as a consequence of physiological adjustments, epigenetics, and/or adaptations to avoid resource limitation[2]. Indeed, fish typically do not have unlimited food resources in their natural environment[107], and may need to balance demand and allocation with availability[98,108]. Larger individuals also have a higher *per-capita* resource demand and may be less capable of obtaining sufficient resources to match warming-enhanced metabolic demands compared with that of smaller individuals[38,106,109]. Specifically, experimental evidence has questioned the ability of fishes to significantly increase absolute energy intake at elevated temperatures, even if available to the individual, due to an inability to ramp up evolutionarily determined maximal predation rates: For instance, predatory fishes which typically deploy a feast and famine approach to foraging[110], may be forced to reduce meal sizes to protect aerobic scope due to the exponential costs associated with digestion of large prey[34,52,111–113], and may be unable to significantly increase capture rates of (smaller) prey[86], thereby placing a natural upper limit to energy acquisition. Indeed, recent empirical work have shown 2–3-fold increases in energy expenditure needed to capture smaller prey and suggested that this bottom-up effect may help explain patterns of shrinking in wild fishes[114,115]. Similarly, continuous grazers such as herbivorous reef fishes typically spend the majority of their waking time foraging under current day temperatures (some more than 80%[116–118]), likely hindering a further 20–40% increase in foraging effort and energy acquisition to match warming-induced metabolic demands.

Our study supports this notion of rising temperature bestowing an advantage for individuals with reduced absolute energetic needs. The fishes examined here all forage on small prey items comprised of fish, crustaceans and other invertebrates[119,120], and diets are not known to differ significantly across adult sizes. That is, a large adult will typically forage on the same nominal prey type and size as a smaller adult but will have to forage faster or longer each day to cover its higher absolute energetic demand. Although larger individuals can have greater tolerance to lack of food[121,122], this is not likely to bestow an advantage if smaller individuals are able to fully cover energetic demands (e.g. for reproduction) where larger individuals cannot, ultimately driving a selection for reductions in maximal size following the ghost-of-limitation-past theory[2]. Supportive evidence has even been observed in the smallest species of coral reef fishes. Based on gut content analyses of Persian/Arabian Gulf and Gulf of Oman conspecifics, Brandl and Johansen et al.[71], attributed size reductions of tiny cryptobenthic fish in the Persian/Arabian Gulf to energetic deficiencies caused by reduced food quality and availability, thus bestowing a competitive advantage to smaller individuals to fulfill absolute energetic demands. Interestingly, the less than 12% subset of Gulf of Oman species that have managed to survive in Persian/Arabian Gulf mirror these ecological commonalities: they are typically species with high habitat and dietary plasticity[73], and able to alter foraging strategies and foraging rates in response to intrinsic and/or extrinsic factors. Indeed, altered foraging patterns of Persian/Arabian Gulf fishes have been documented both in-situ and in-vivo, with e.g., *Pomacanthus maculosus* (a typical herbivore/omnivore[94,123,124]) feeding predominantly on coral and sponges in Persian/Arabian Gulf[94], while *Pomacentrus trichourus* (a typical planktivore[125]) shift to benthic resources during the Persian/Arabian Gulf summer[94,95]. Importantly, these patterns have developed across numerous generations within the >6,000-year history of the Persian/Arabian Gulf, highlighting that observed patterns are not likely to be short term trade-offs. Additionally, this balance between absolute resource supply and demand is linked to external ecological conditions such as ecosystem productivity and resource availability, and is therefore different to the largely intrinsic oxygen supply hypotheses discussed above.

## Consolidating theory

Our data supports a modified consolidated energy acquisition theory for warming induced size reductions in fishes. It is undoubtedly true that in order to survive at elevated temperatures a species must possess adequate aerobic surplus to fuel critical functions such as swimming, digestion, and reproduction[21,34]. As such, we lean on the theoretical underpinnings of both GOL and MASROS to state the first essential tenet of survival is to "Protect Aerobic Scope" (PAS; see also Jutfelt et al.[34], for similar conclusions). For species that overcome the PAS hurdle, either through existing species-wide capacity or population-level adaptation, the greater absolute energy acquisition required by larger individuals may quickly become a secondary hindrance for fitness and survival. Many aquatic ecosystems are expected to become less, not more, productive in warmer oceans[126–128] and the *per-capita* ability of organisms to capture, process and assimilate adequate food are therefore likely to decline at elevated temperatures[129–131]. Ecological metabolic theory and resource limitation theory both suggest that unless energetic demand is met with adequate resource acquisition, then growth, development, and survival will be compromised as diminished resources are necessarily reallocated to fitness enhancing processes such as reproduction[107,132]. Such patterns have been seen in a myriad of organisms ranging from plants and insects to ectothermic and endothermic vertebrates[107,132–134], including recent direct experimental evidence[98]. However, theoretically this temperature-driven increased demand for resources could equally be counteracted by reducing the *per-capita* energetic demand

to match the natural upper limit to energy acquisition for a given species.

Using established equations for temperature sensitivity of fish metabolism, a 3.0 °C increase in ocean temperature is expected to incur a 16–30% increase in basal energy demand based on a $Q_{10}$ of 1.65–2.40[41,48,91–93]. Simultaneously, most fishes are thought to have a metabolic mass-scaling exponent of -0.8–0.9 (e.g. ln(SMR) = 0.8 * ln(mass) – 5.43[48]), and our data confirms those expectations with an average exponent of 0.9 ± 0.05 (±S.E.M., see Fig. 5). As a result, any given species could offset the expected increase in absolute energy demand by reducing mass by -18–28%. Accordingly, Persian/Arabian Gulf cryptobenthic fishes are known to be 9.1–40.2% smaller in mass than Gulf of Oman conspecific fishes[71] and larger bodied species such as *L. ehrenbergii* and *Pomacanthus maculosus* are known to be ~32–46% smaller in Persian/Arabian Gulf (based on published TL reductions of maximum body size and weight-length relationships[77,135,136]). Consequently, the observed size reductions in Persian/Arabian Gulf fishes, which match the 14–40% size reductions projected globally, theoretically all but eliminates the need to increase absolute energy acquisition above the evolved upper limit for prey capture and digestion for any given species (Fig. 6).

Although undoubtedly more complex than presented here, our modified theory leans on a combination of established resource limitation theory (e.g. supply–demand hypothesis and resource-allocation theory[107]) and metabolic theory[41] to state that the second essential tenet of survival at elevated temperatures is to Limit Energy Demand (LED) relative to the upper limit for prey capture, digestion, and energy assimilation (i.e. maximal energy acquisition). Our full theory of "Protect Aerobic Scope and Limit Energy Demand" (PASLED), thus encompasses the patterns of performance and size reductions observed in Persian/Arabian Gulf fishes, including protection of performance metrics needed to maintain prey capture and digestive processes[34]. Importantly, although additional supportive research is needed, PASLED theoretically also allows for species-specific projections of size reductions based on evaluations of aerobic surplus and the capacity to maintain, increase, or alter energy acquisition (e.g. via altered prey consumption) to support rising energetic demands[50,137,138], which should cause predictable disparities among species and feeding modes. For instance, large fish such as sunfish (*Mola mola*, Linnaeus 1758), devil ray (*Mobula diabolus*, Shaw 1804), and whalesharks (*Rhincodon typus*, Smith 1828) all exist in tropical and subtropical waters showing that large species can adapt to life at high temperatures. While neither GOL or MASROS can fully explain these occurrences[31,34], nor why some species appear to shrink more than others as temperatures rise[24], our PASLED theory suggest that additional insights can be gained by comparing metabolic demands with maximal species-specific prey and energy acquisition (see Box 1).

## Summations

Our data reveal two evolutionary pathways for survival of tropical reef fishes at elevated temperatures. Survival based on an evolutionary history that has bestowed a high inherent thermal tolerance and the ability to perform regardless of present-day temperature rises, and survival based on rapid adaptation of thermal performance windows. Our results thus confirm that although low adaptive capacity is expected for tropical reef fishes, some species do appear to have that capacity. However, given the fact that less than 12% of the species found in Gulf of Oman have managed to survive in the adjacent and biogeographically connected Persian/Arabian Gulf[76,139], it is unlikely that adequate adaptive capacity will be seen across a broad range of tropical reef fishes. Additionally, the current pace of climate change (+3.0 °C within 50–100-years) far outpaces the 6000 years of elevated temperatures within the Persian/Arabian Gulf, further limiting the

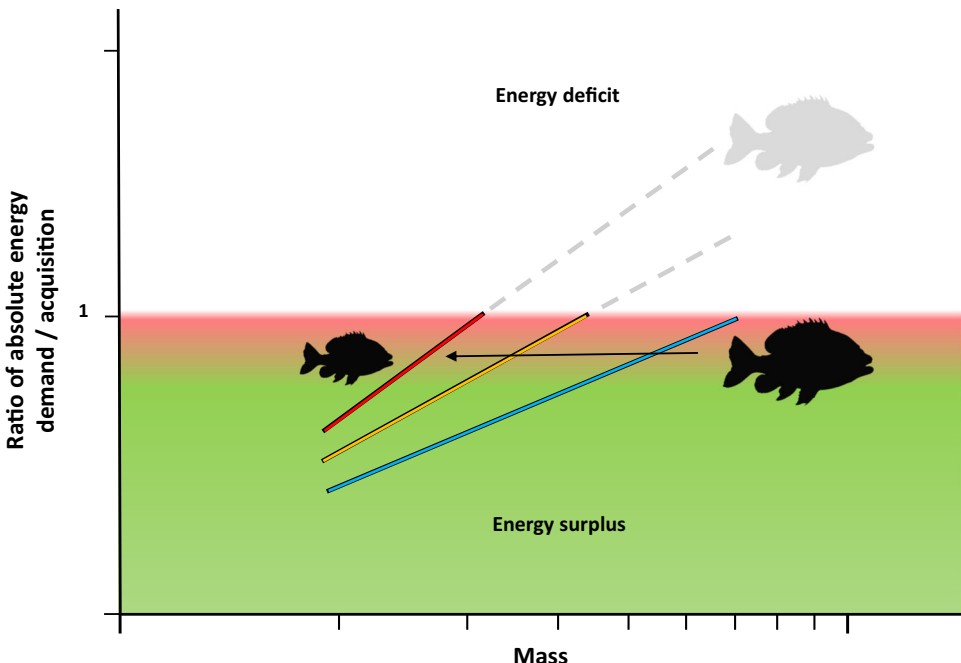

**Fig. 6 | Theoretical depiction of the Protect Aerobic Scope and Limit Energy Demand (PASLED) theory as it relates to absolute energy demand and acquisition across mass.** Absolute energy demand is defined as the total daily calories needed for survival by a given individual, including species-specific needs for critical ecological functions (e.g., swimming to forage, reproduce etc.). Maximal absolute energy acquisition is defined as the maximal calories acquired from the combination of prey capture, digestion and assimilation. X-axis shows mass, while Y-axis shows the ratio of absolute energetic demand to absolute energy acquisition. Values < 1.0 (green areas of graph below red zone) highlight conditions where energetic demand is lower than maximal energy acquisition, signifying an energy surplus. Values > 1.0 (denoted by white areas of graph above red zone) highlight conditions where maximal energy acquisition is insufficient to fulfill demand, signifying an energy deficit. The red transition zone signifies the onset of mitigation, where plastic responses may be expected, e.g., dietary changes. As temperatures increase (depicted as blue, orange and red lines, respectively), larger individuals will eventually require more absolute energy than they can acquire consistently on a daily basis. Arrow depicts the expected reduction in maximal mass at elevated temperatures. Note that the width of the red transition zone may change across mass, and among species and feeding modes.

## BOX 1:
# PASLED energy acquisition hypotheses and predictions

PASLED hypotheses:
1. For species exploiting the same, easily digestible, prey sizes and types across adult sizes, e.g., whalesharks[116–118], we hypothesize size reductions to depend primarily on energy acquisition from maximal sustained *per-capita* prey capture rates.
2. For species exploiting larger or different prey as larger adults, e.g., gape limited prey selection in piscivores[170], we hypothesize size reductions to depend on energy acquisition from the combined effect of maximal *per-capita* capture rates and digestion (rates and efficiency) across all available prey sizes (as applied by Jutfelt et al.[34]).
3. For omnivorous species, we hypothesize size reductions to depend on maximal sustained *per-capita* energy acquisition when exploiting the broadest possible array of prey items in the ecosystem (including prey not typically exploited in cooler systems as shown by Shraim et al.[94]).

PASLED predictions: As temperatures rise above optimum, we predict:
a. Maintained maximal foraging rates in species utilizing (near) continuous foraging modes on small prey items, e.g., among herbivores;
b. Reduced meal sizes and (if possible) increased number of meals in species typically relying on feast-and-famine foraging and increasingly larger prey items as larger adults, e.g. among piscivores;
c. A broader and more opportunistic diet in omnivorous species, including increased reliance on easily captured and/or lower nutritious prey;
d. Reductions in maximal size of fishes even when provided with unlimited food supply.

likelihood of adaptation. For those that do survive, size reductions appear to be a universal consequence of ocean warming, but our findings show that the Temperature Size Rule puzzle will remain unresolved if we keep attempting to explain the complexity of growth simply as a consequence of oxygen supply limitations. We suggest that the balance between energy demand and acquisition, are primary drivers of the Temperature Size Rule and shrinking body size of fishes in warming waters, and our consolidated PASLED theory encompass these ideas. PASLED not only provides plausible and testable hypotheses (including why some species appear to shrink more than others) but also highlights additional research needs, including likely changes to prey types, prey nutritional values and prey abundance, as these

parameters will directly impact fish numbers and sizes. While the consequences of ocean warming remain uncertain for most species, this study reveals a path forward to increase our understanding of species-specific responses and, ultimately, our capacity to project warm-induced changes to fish and fisheries globally.

## Methods

All collections and trials were conducted under NYUAD animal ethics permit IACUC 17-0002, 20-0001, and Environment Abu Dhabi, UAE collection permit TMBS/17/l/284.

### Field locations

Exploiting geographic regions with drastically different annual sea surface temperature (SST) profiles, this study was conducted across six reef sites in the southeastern region of the Persian/Arabian Gulf and northwestern region of the Gulf of Oman. The chosen Persian/Arabian Gulf sites reach typical summer maximum SSTs of 36.0 °C (reef sites: Dhabiya: 24.36383°, 54.10121°; Ras Ghanada: 24.84743°, 54.69235°; Saadiyat: 24.65771°, 54.48691°), whereas the chosen Gulf of Oman sites are more typical of coral reef temperature profiles globally, reaching a summer maximum of 32.0 °C (reef sites: Dibba Rock: 25.55378°, 56.35694°; Sharm Rock: 25.48229°, 56.36695°; Snoopy Rock: 25.49210°, 56.36401°, Fig. 7). These summer extremes usually persist within 2.0 °C of maximum from late May until September[73,79,140].

### Study species and collections

Reef fishes from two families were selected as good indicators of thermal adaptation potential in reef fishes. The species *Lutjanus ehrenbergii* and *Scolopsis ghanam* were chosen for study since they are among the most abundant fishes in Persian/Arabian Gulf and Gulf of Oman while also representing evolutionary distinct lineages (Lutjanidae and Nemipteridae[73]). In addition, the chosen species are piscivorous/omnivorous[119,120] and are by way of feeding mode forced to swim when foraging. By comparing individuals from three Persian/Arabian Gulf and three Gulf of Oman sites representing the divergent thermal environments, these fishes provided the opportunity to evaluate consequences of elevated temperatures on a range of performance metrics within closely related populations (intraspecies) as well as across evolutionary traits (interspecies).

From January 2018 to July 2020, a total of 85 fishes (*Lutjanus*: $n = 44$, $12.3 \pm 0.3$ cm standard length (SL), range 9.2–16.5 cm SL, $48.1 \pm 3.36$ g, range 20.0–116.1 g; *Scolopsis*: $n = 41$, $11.5 \pm 0.2$ cm SL, range 9.2–14.2 cm, $40.2 \pm 2.4$ g, range 16.6–87.4 g; mean ± S.E.M.) were collected by scuba divers using fine-mesh monofilament barrier nets. Only adult fish were used in this study to avoid ontogenetic differences, and collected size ranges were matched across Persian/Arabian Gulf and Gulf of Oman to the greatest extent possible. All collections were conducted during periods of comparable ambient water temperatures (i.e. when SST was $27.0 \pm 0.5$ °C, $31.5 \pm 0.5$ °C, $35.5 \pm 0.5$ °C in the Persian/Arabian Gulf region; and $27.0 \pm 0.5$ °C, $31.5 \pm 0.5$ °C in the Gulf of Oman region). These temperatures corresponded to the approximate annual mean and summer max temperatures found in Persian/Arabian Gulf (27.0 and 35.5 °C) and Gulf of Oman (27.0 and 31.5 °C) regions[71], and allowed collections and corresponding trials to occur during the shoulder seasons and summer.

After collection, fishes were transported to the seawater laboratory facilities at New York University Abu Dhabi (NYUAD) and held in $80 \times 40 \times 40$ cm tanks (length × width × height) under a 12–12 h light-dark regime (subjected to sunrise as beginning of daylight). Fish were housed in groups of four fish per tank and tagged with visible elastomer tags (Northwest Marine Technology, Inc.) to allow individual identification. Tanks were continuously supplied with filtered seawater from discrete sumps (containing a protein skimmer and canister filter to maintain water quality), at the ambient collection temperature of 27.0, 31.5, or 35.5 °C (mean ± 0.1 °C) and 40ppt salinity, equivalent to

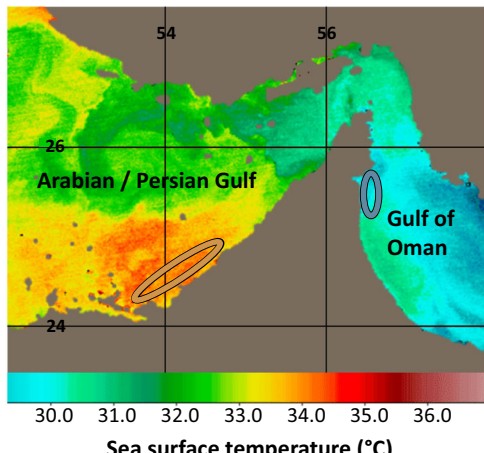

**Fig. 7 | Heat map of maximal sea surface temperatures in the Persian/Arabian Gulf and Gulf of Oman.** Map shows latitudes and longitudes as well as study locations in dark orange and blue circles, respectively. Sea surface heat map adapted with permission from Yao and Johns[69].

the mean ppt in Persian/Arabian Gulf and Gulf of Oman sites[79,141]. All fish were fed twice daily to satiation with commercial fish foods and left undisturbed to settle to lab conditions until regular feeding patterns were observed (7–10 days).

As fish were field-collected across seasons, all individuals were assumed to be naturally acclimatized to their collection temperature (i.e., 27.0, 31.5 and 35.5 °C). However, only Persian/Arabian Gulf fishes could be collected at 35.5 °C. In order to compare the relative capacity of Persian/Arabian Gulf and Gulf of Oman fishes to function in 35.5 °C water as an indication of thermal adaptation which have occurred explicitly in the Persian/Arabian Gulf population, Gulf of Oman fishes were collected during peak summer at $31.5 \pm 0.5$ °C and acclimated for three weeks to $35.5 \pm 0.5$ °C to maximize acclimation potential (following Johansen et al.[142]). Specifically, once fish had settled to lab conditions at ambient collection temperature (i.e., 31.5 °C) temperatures were raised by 0.5 °C/day. Once at 35.5 °C, fishes were left to acclimate undisturbed for an additional 21 days. Persian/Arabian Gulf 35.5 °C fishes were held under laboratory conditions for the same time period to ensure comparable conditions were met. The three-week acclimation period was chosen because previous research has demonstrated that maximal adjustments of metabolic metrics will usually occur within a three-week period in tropical reef fish (sensu Johansen et al.[142]). Acclimation was stopped for individuals showing signs of distress, which included all individuals of Gulf of Oman *S. ghanam* at 35.5 °C.

All fish were left unfed for 24 h before experimental trials commenced to ensure a postabsorptive state that maximized the energy available for swimming[143], and all trials were conducted within the 12 h daily light regime in order to match the diurnal activities of the study species. Following ethical requirements to limit animal numbers for research, after trials were completed, all fish were humanely euthanized to allow collection of ecologically and physiologically relevant tissues for additional studies unrelated to this publication (including gonads, otoliths, hearts, gills, liver, muscle).

### Metrics of performance

A total of 10 performance metrics (defined in the following) were compared across regions, including four metrics of cardio-respiratory metabolic performance (SMR, MMR, aerobic scope, and COT), three metrics of kinematic muscle performance (tail beat Frequency, Amplitude and Strouhal number), and three metrics of swimming capacity (optimum, burst, and critical swimming speed; $U_{opt}$, $U_{burst}$,

$U_{crit}$, respectively). Specifically, the metabolic performance of individuals can be described on the basis of SMR, which show the minimal oxygen supply needed to maintain basal bodily functions; MMR, which show the maximal oxygen supply capacity; aerobic scope, which highlights the oxygen supply beyond SMR that is available for critical functions such as swimming; and COT, which denotes the lowest achievable energetic cost per distance traveled and which is critically important for e.g., migration. For kinematic muscle performance, fish obtain maximum thrust per unit input energy at the tail beat Frequency of maximum Amplitude[144]. Strouhal combines tail beat Frequency and Amplitude to provide a dimensionless estimate of this propulsive efficiency[61,145], usually ranging from 0.2 to 0.4 in swimming fish, with higher values indicating lower propulsive swimming efficiency[146]. Finally, for swimming performance, $U_{opt}$ is defined as the swimming speed which provides the lowest energetic cost per distance traveled (i.e. swimming speed at COT); $U_{burst}$ signifies the swimming speed which causes a change in swimming mode from steady propulsion to an onset of recruitment of white anaerobic muscle for burst propulsion and a gradual build-up of oxygen debt[147]; while $U_{crit}$ reflects the maximum prolonged critical swimming speed of a species, which dictates its capacity for chasing prey and evading predators[55,148]. When combined, these 10-performance metrics provide a comprehensive evaluation of 1) the relative capacity of Persian/Arabian Gulf and Gulf of Oman fishes to maintain critical ecological functions across thermal gradients, 2) whether energetic efficiency differs between environments, and 3) whether larger individuals are disproportionately impaired at elevated temperatures as predicted by GOLT and MASROS.

## Swimming respirometry and kinematics

At each temperature the oxygen uptake and swimming ability (mode, speed and kinematics) of each species was quantified for 6–9 individuals swimming solitarily in a 10L clear Plexiglas swim-tunnel respirometer. Two identical Steffensen-type respirometers were used simultaneously, each with a working section of $8.0 \times 8.0 \times 10.0$ cm (length × width × depth). Individuals were assigned to each respirometer at random and were unable to see one another. Flow within the working section of the respirometers was calibrated from 0 to $125.0 \pm 0.5$ cm s$^{-1}$ (mean ± SE) using a digital TAD W30 flow-meter (Hoentzsch, Germany). Solid blocking effects of the fish in the working section were corrected following Bell and Terhune[149] and were kept below 5%.

At the beginning of each trial, a respirometer was filled with temperature controlled (27.0, 31.5 or $35.5 \pm 0.1$ °C, mean ± SE), filtered and fully aerated seawater. Next, a fish was placed in the respirometer and left to acclimatize for ~8 h at a swimming speed of 0.5 body length per second average (bl s$^{-1}$), until oxygen uptake of the test subject reached a steady state level and the fish had settled into a continuous slow swimming rhythm[4,55,150,151]. The trial was then started and the oxygen uptake of the test subject was measured at increasing swimming speeds for a total of 30 min at every speed, using 0.6–1.4 bls$^{-1}$ speed increments. The maximum swimming speed tested was dependent on the swimming ability of the individual fish, with flow velocities incrementally increasing until the fish could no longer keep up and was swept downstream with the flow onto a retaining grid for longer than 5 s. At this point the flow velocity and total swimming time was recorded, and the fish returned to a 0.5 bls$^{-1}$ swimming speed to recover. Once oxygen uptake reduced to a steady state level (within 20% of the oxygen uptake at beginning of the experiment), the fish was deemed to have recovered, the experiment was stopped and the fish was returned to its holding tank (usually within 30–120 min, see Supplementary Fig. 1 for details). During the trial, the fish was continuously monitored for swimming mode (i.e., steady versus burst swimming), and the total swimming time and flow speed was recorded at the point of change from steady caudal swimming to burst-and-coast propulsion

for longer than 5 s continuously. Gait change was determined because steady aerobic caudal swimming is the swimming mode most commonly used by these reef fish for routine tasks such as foraging, whereas burst-and-coast propulsion signifies the onset of partial recruitment of white anaerobic muscle for propulsion that causes a rapid build-up of oxygen debt and sets a short time limit to activities such as predator evasion[152].

For every oxygen measurement, a dynamic ~240 s flush, ~60 s equilibration and ~300 s measurement period was applied following the intermittent flow respirometry methodology of Steffensen et al.[153], Steffensen[154] and Svendsen et al.[155]. The flushing period ensured the oxygen concentration throughout the trial did not decrease below 80% of air saturation and reduced any $CO_2$ build up. Oxygen levels within the swimming respirometers were measured using fiber optic oxygen meters and monitored with AutoResp V2 (Loligo Systems, Denmark) and AquaResp V3[155] (Pyroscience Firesting sensors, Germany). To reduce bacterial growth and respiration within the system, the respirometers were routinely treated with a 2% chlorox solution and thoroughly flushed with freshwater. This procedure ensured average background respiration remained below 10% of the oxygen consumed by fish during swimming trials. In particular, immediately prior to and following the completion of each trial the respirometers were run for one additional 30 min cycle at 1.2 bl s$^{-1}$ during which the oxygen uptake of the empty respirometers was measured. We subtracted the proportional background respiration from each oxygen uptake measurement following Rummer et al.[156].

## Data extrapolation

**Swimming energetics and performance.** $U_{burst}$ and $U_{crit}$ were calculated following Brett[57]: $U_{burst \; or \; crit} = U_p + U_i \times (t/t_i)$, with $U_p =$ the penultimate flow speed before the fish changed gait from strictly caudal to burst-and-coast swimming for >5s ($U_{burst}$), or before the fish fatigued (i.e. stopped swimming) and was pinned to the downstream grid for >5s ($U_{crit}$), $U_i =$ each swimming speed increment (in bl s$^{-1}$), $t =$ the length of time in the final increment where gait change or fatigue occurred, and $t_i =$ the set time interval of each swimming speed increment (30 min). COT was calculated as the minimum oxygen demand required to swim 1 km (following Claireaux et al.[157]) based on a reciprocal quadratic regression ($y = 1/(a + bx + cx^2)$), with y = Cost of transport (COT) in mgO$_2$/km and x = swimming speed in cm/s. Temperature sensitivity of fish metabolism (i.e. the rate of increase in response to increasing temperature) was calculated as $Q_{10} = \left(\frac{R2}{R1}\right)^{\left(\frac{10}{(t2-t1)}\right)}$.

**Size correction.** In accordance with climate change projections for shrinking body size of fishes in warmer waters[17–20,24], adult Persian/Arabian Gulf fishes are consistently smaller than Gulf of Oman fishes of the same age[71,77]. Despite significant effort to collect a similar range of sizes from both regions, collected Persian/Arabian Gulf fishes were generally smaller in mass (L. ehrenbergii $37.6 \pm 4.3$ g versus $62.8 \pm 3.4$ g; S ghanam $34.3 \pm 1.8$ g versus $48.4 \pm 4.8$ g) and standard length (SL, L. ehrenbergii $11.5 \pm 0.3$ cm versus $13.6 \pm 0.3$ cm; S ghanam $10.9 \pm 0.2$ cm versus $12.1 \pm 0.4$ cm, mean ± S.E.M.), but were equal within treatments for each region (Supplementary Data 4). Additionally, for the 10 examined performance metrics there is currently no consensus for the relationship between size and temperature[47,53,158]. Thus, in order to avoid dictating the combined effect of body size and temperature and to facilitate a balanced comparison of Persian/Arabian Gulf versus Gulf of Oman performance within and across temperatures that were unaffected by regional size differences, data were size adjusted in R using four automated steps (see r-script and raw data in Supplementary Data 1): 1) total length or mass was removed from each performance measure ($U_{opt}$, $U_{burst}$, $U_{crit}$, Frequency, Amplitude, SMR, MMR, aerobic scope, COT) to obtain the raw data (e.g. data was converted from bl/s to cm/s and from mgO$_2$/kg/hr to mgO$_2$/hr); 2) a linear

regression between the Log (raw data) of each performance measure and the Log (mean size across temperatures and regions) was then used to calculate the residual deviation of each individual performance measure from the expected value; 3) the effect of size was then removed by adding the residual for each individual performance measure to the expected performance of a mean sized individual; 4) Finally, for visual representation, size adjusted raw data was back-transformed to size-specific data using means. As we were explicitly interested in comparing responses that were conditional on the body-size covariate, we opted to use this baseline covariate adjustment approach to statistically remove its effects from our models[159], as this approach has been shown to control against imbalances in covariates and increase statistical power in randomized trials by reducing the number of model factors, by reducing model complexity, and by increasing available degrees of freedom[159–164]. Only Strouhal number was not scaled as it is a dimensionless parameter.

**Swimming energetics and performance.** Measures of swimming performance and oxygen uptake were plotted as swimming speed (bl s$^{-1}$) on the $x$-axis and mass corrected oxygen uptake (mg O$_2$ kg$^{-1}$ h$^{-1}$, MO$_2$) on the $y$-axis. Standard metabolic rate (i.e. SMR, oxygen uptake at rest) was then extrapolated from a nonlinear regression of oxygen uptake measures following $y = a + be^{cx}$ (with SMR as the $y$-axis intercept[150], see Supplementary Fig. 1). MMR was taken as the highest recorded oxygen uptake across a single measuring period[150,153–156], which always occurred either at or immediately before $U_{crit}$. Aerobic scope was calculated as MMR-SMR.

**Kinematic muscle performance.** At each swimming speed, a set of three 60 s videos of the fish was recorded for subsequent kinematic analyses of tail beat frequency, amplitude and Strouhal number. All videos were recorded simultaneously from above and the side at 100 fps, using a single high-speed camera (Kayeton Ltd) in 1280 × 720 resolution. The side view was captured using a 45degree mirror placed next to the swim section, thereby allowing the top and side view to be captured by a single camera. For each video, five consecutive tail beats of steady-state swimming were analyzed for Freq and Amp using LoggerPro 3.14.1. The mean tail beat Frequency and Amplitude for each fish at each velocity were then used to calculate Strouhal number as (Frequncy x Amplitude)/swimming velocity.

### Statistical analyses
**Swimming energetics and performance.** For each species, individual differences in metabolic and swimming performance metrics were compared between the test temperatures and regions (i.e., Persian/Arabian Gulf versus Gulf of Oman) using a linear mixed effect model with temperature, region and metric as the fixed factors. Individual ID was included as a random effect to account for multiple data points stemming from the same individual. This was followed by a planned comparison for least squares means for specific differences across temperatures and regions within each metric of interest.

**Kinematic muscle performance.** We compared the slopes of Frequency, Amplitude and Strouhal data against swimming speed using a linear mixed effect model for each metric and species, followed by least squares trends within and across temperatures and regions. Each linear mixed model included temperature and region as fixed effects, swimming speed as continuous covariate and individuals ID as a random effect. Where region trends did not differ within temperatures, data were pooled to increase statistical power, followed by post-hoc planned comparisons.

**Size effect of temperature.** To test the merits of the GOL and MASROS predictions, we compared the slopes of the raw aerobic scope, $U_{crit}$,

SMR and COT data against mass using a linear mixed effect model for each metric and species, followed by a least squares trend for estimating and comparing mass-scaling slopes within and across temperatures and regions. Each mixed model included temperature and region as nested random effects, and mass as a continuous covariate to account for sampling variance. Where region trends did not differ within temperatures, data were pooled to increase statistical power, followed by post-hoc planned comparisons.

For all analyses, model fits were examined using the Akaike information criterion (AIC), calculations of marginal and conditional R$^2$, and partial effect sizes for mixed effects models. In linear mixed models, there is no current consensus on how to accurately estimate all sum-of-squares (SSs) needed to estimate the percent variance explained by a model term. However, partial SSs, marginal and conditional (semi-partial) R$^2$, and partial effect sizes can be approximated. Marginal R$^2$ describes the variance explained by the fixed effects, whereas conditional R$^2$ describes the variance explained by the full mixed model[165]. Partial effect size describes the percent of the partial variance associated with a single term after accounting for other predictors in the model[166], and is derived as Eta Squares, which simulates the effect size in a design where only the term of interest was manipulated[167]. To avoid inherent positive bias in eta square estimates, Adjusted Partial Eta Squared (Adj η$^2$p) provides unbiased population estimates by using unbiased measures of the variance components[166,168]. All model data were tested for univariate assumptions using Shapiro-Wilk normality test, Levene's test for homogeneity of variance and Grubb's outlier test, which define outliers based on the largest absolute deviation from the sample mean. Evidence of overdispersion was tested as the ratio of residual deviance to degrees of freedom from the summary model outputs, with values above 1 indicative of overdispersion. Data that did not initially comply with assumptions were Box-Cox transformed and a total of two single data points were removed as outliers (out of >1800 data points). As *L.enhrenbergii* displayed unsteady swimming at speeds ≤1.9 bls and *S. ghanam* at ≤1.1 bls, species-specific kinematics could not be accurately evaluated at these speeds and were excluded from analyses. All final model data met assumptions. For all comparisons, False detection rate was used to correct for Type I errors[169]. All data were analyzed using the packages 'lme4', 'lmerTest', 'MuMIn', 'multcomp', 'languageR', 'LMERConvenienceFunctions', 'emmeans', 'EnvStats', 'outliers', 'effectsize', 'r2glmm' and 'car' in R 4.2.2, while graphs were completed using SigmaPlot V.14.

For a statistically detailed rendition of findings, see Supplementary Results.

### Reporting summary
Further information on research design is available in the Nature Portfolio Reporting Summary linked to this article.

## Data availability
All source data and analytical codes are provided in the Supplementary Data 1 file with this publication. This file contains two directories, one for primary statistical analyses and one for secondary statistical analyses. The primary stats folder contains all data and analyses included in the main manuscript, which utilize mass adjustment before comparisons of performance metrics. The secondary stats folder contains all data and analyses needed to conduct the same analyses with mass inclusion as a covariate instead of mass adjustments. This latter method produces near identical results to the main manuscript data but with lower statistical power due to the added model complexity.

## Code availability
All analytical codes are provided in the Supplementary Data 1 file with this publication.

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

## Acknowledgements

This work was supported by Mubadala ACCESS grant CG009 from Tamkeen and grant AD004 from NYUAD to J.A.B., and the Biotechnology and Biological Sciences Research Council Doctoral Training Studentship to D.M.R. Their funding is greatly appreciated. We thank the NYU Abu Dhabi Core Technology Platforms' Marine Research unit for support of field collections. We further thank D. McParland and N. Al-Mansoori for field support.

## Author contributions

J.L.J., G.O.V., J.A.B and H.A.S. designed the study; J.L.J., M.D.M., and G.O.V. performed field collections; J.L.J., M.D.M., and D.M.R. ran physiological trials; J.L.J., M.D.M., G.O.V., and D.M.R. performed laboratory work; J.A.B. and H.A.S. provided funding and resources; J.L.J. and D.M.R. performed data analysis and visualization; J.L.J. wrote the first draft of the manuscript, and all authors contributed to writing thereafter.

## Competing interests

The authors declare no competing interests.
