## [Peer Review File · Nature Communications]

Shrinking fishes on the world's hottest coral reefs - cause and consequences of surviving ocean warmingReviewers' Comments:

Reviewer #1:

Remarks to the Author:

This study examines 10 metabolic and physiological performance metrics in two reef fish species living in Gulf of Oman (GO) and Arabian Gulf (AG), aiming to assess whether AG populations, typically exposed to extreme summer temperatures (35C) have developed adaptations to improve their performance. The authors aim to test two oxygen limitation theories (gill oxygen limitation and MASROS – maintain aerobic scope) as a potential explanation for decreasing maximum body sizes of fishes in warmer temperatures. They find limited support for either of the two oxygen limitation theories and instead suggest an alternative theory based on limitation of food availability.

Overall, I found that that the questions, experimental design and execution are generally well performed. It is clear that a lot of work was put into conducting these experiments and the question is generally interesting to the wide audience. This study could become an important paper, but it needs significant work on the presentation of results and discussion.

My major comments are:

1) Data availability: First and most importantly, I could not find the data available for review, so could not play with the results and analyses and see whether they make sense to me. I could see that the code (in a Word file!) was actually provided in the supplementary material, but without the original data and in this kind of .docx file it is useless. It is essential that for the next round of review the full data and code are made available to reviewers. It is easy to do that by zipping the entire R project and uploading it for review.

2) Mixed effect model structure: I am not sure why the authors have used such a convoluted four step method to account for covariation of physiological metrics with body size. Without having the data to assess these steps it was impossible to understand whether this method worked. However, the authors should redo the analyses and include body size in main mixed effect models as a numeric fixed effect. Then you can check for interactions between body size and temperature (i.e. – is scaling different at different temperatures) and regions. Once body size is included in the main analyses you do not need to run separate and confusing analyses regressing raw data against body size. Moreover, it is not clear why temperature is treated as a random effect in these additional body size models (random effects should have at least 5 levels of data and you have 2 or 3, depending on the species). As you are citing Wootton et al. study, I suggest looking at their statistical analyses, where physiological measures are tested across temperatures and where body mass is included as a covariate in the models. The data and R code used for the analyses are publicly available. Second, I am not sure why the individual ID is included as a random effect in the model. Is this to account for overdispersion or were the repeated measures taken for each individual for a specific metric? Was there evidence for overdispersion and was that tested? It was not entirely clear to me whether fish were individually tagged and separated for the experiments.

3) Use of 35.5C temperature without long-term acclimation. The GO individuals do not experience 35.5C temperatures. In this study they were acclimated to such temperatures for 21 days, but this is absolutely not enough for the question of long-term acclimation and adaptation. The fact that one species showed high mortality (100% mortality?) just confirms that. Again, as you refer to Wootton et al. study, they showed that 3 generations of acclimation may be needed before SMR is fully acclimated to new temperatures. I understand that 3 generations may not be feasible for many studies, but we should not interpret short-term acclimation studies in the long-term adaptation context then. I strongly suggest focusing main results on 27 and 31.5C temperatures and including 35.5 just as additional special case for the AG species only (because they are naturally acclimated to these conditions).

4) Presentation of results. To be honest, current presentation of results is too hard to read and comprehend. It mechanically reports various statistics, which should be instead placed in a table. Each section is repetitive and the reader quickly loses track of what is found and why. Since you present quite a lot of different results, it is essential that results are presented clearly, visual summaries are informative and repetitive statistics are reported in a table. In addition to or instead of performing multiple planned comparisons, please include body weight in the main model and use mixed effect

model summaries and plots of highlight statistical model outputs in a more intuitive way. You can use packages and functions "effects", "predict" and show model predicted responses with uncertainty, with raw data in the background. This way you can show variation in the actual raw data and various statistics predicted by the model (you can use the publicly available code from Wootton et al for the plots, if it is useful). Please move all the statistical reporting from the Results text into tables and figures, and instead present results in plain language, indicating main points, your hypotheses and main findings.

5) Introduction of a new hypothesis for temperature effect on maximum body size. Your study has great results about various physiological metrics across temperatures, but you did not measure resource availability and usage. For me your main result is that SMR across 4.5C temperature difference (ie between 27 and 31.5C) is generally the same, which is an important and interesting finding. But it does not fit your subsequent speculation about resource limitation driven by temperature. There is no data in this study to support it and it is based on wrong information. For example, you do not present data on maximum body sizes in this study. Also to justify resource limitation ideas, you assume Q10 of 2-3 and 23-39% increase in basal energy demand across 3C temperature difference (line 647). But your own data shows that for SMR (i.e. basal energy needs) Q10 is 1! There was no difference across 4.5C in temperature. So why should basal energy demand increase? Also, two theories on oxygen limitation (line 69) are not the only theories aiming to explain why body sizes are smaller in warmer temperatures. The main competing theory is life-history based explanation, again presented in Wootton et al., but not mentioned in your study. It suggests that faster juvenile growth and earlier maturation diverts resources to reproduction instead of growth (either for adaptive or some limiting reasons, which is a separate and complicated question). Moreover, there are already multiple hypotheses suggesting resource limitation as an explanation for smaller body sizes in warm temperature. See Verberk et al. 2021 and Audzijonyte et al. 2019 for reviews on these theories. Therefore, I strongly believe that the speculative part of "resource limitation" and "consolidating theory" should be completely removed from this paper. It fits better for a separate paper, if the authors wish to pursue it. Focus on your interesting results and interpretations, such as very little difference in performance rates across 4.5C temperatures (assuming this result remains unchanged once body mass is included as a covariate in the main model), different degrees of adaptation in the two species, better discussion on swimming performance metrics. Please make sure you summarise your key findings in the discussion again and discuss them specifically, comparing your findings with other studies. Currently, the discussion is too general and not relevant to your findings.

6) Comparison of GOLT and MASROS hypotheses. GOLT has been misinterpreted and misunderstood by many researchers. I am not a big fan of it myself, but it is important to interpret it correctly. It is not very different from MASROS, because it suggests that maturation is triggered by decreasing oxygen availability in the blood (due to different scaling of gill surface area and body volume) and further growth is reduced. It does not say that large fish do not have aerobic capacity left, because obviously that would be un-adaptive and a silly thing to do. So both hypotheses are talking about the overall maintenance of aerobic scope and I am not sure why they are treated differently.

Minor comments:

- It is not really clear that the maximum size of many species is reducing due to temperature. The study you cite in support of this (line 107) is Cheung et al. 2013 and it does not use any empirical data for their claims.
- On line 470 you conclude that *S. ghanam* was partially limited by mass, but it is not clear why this conclusion has been reached, as COT increased with mass at all temperatures. It is essential not to add ad hoc interpretations to our results. Therefore please include mass in the main model and test for interactions between mass and temperature and region. This way you can discuss whether there was any significant interaction. It is still not clear what such interaction would actually mean in terms of limitations. In order to show that e.g. fishes oxygen availability become more limited in larger sizes we would need to show that the ABILITY to uptake oxygen does not scale as fast as the demand for oxygen. I am not sure whether anyone has done that.
- On line 561 – Wootton et al. did not say that existing GOLT evidence comes from short term studies,

they did not discuss existing evidence for GOLT at all. They said that increased energetic or oxygen demands in warmer water are shown mostly in short term studies, yet when they allowed for multi-generational acclimation, they found that there is no difference in SMR between 4C temperature. This is also largely your finding, yet for some reason you still assume that higher temperatures will increase energetic demands.

Reviewer #2:

Remarks to the Author:

The system the authors have used in this study is unique and perfect for empirically testing theoretical hypotheses regarding shrinking in fish with increasing temperatures. Overall, I found the paper interesting and adds much needed data to this research area. I do however have some concerns regarding the experimental design and interpretation of the results which could use some clarification (see detailed comments below).

The PASLED hypothesis the authors present at the end of the discussion is a compelling one with clearly defined and easily testable predictions, it has foundations in various previous hypotheses but uses the ideas in a new way to predict how reductions in body size will vary depending on the species foraging mode and prey type. Whether or not it should be included in the discussion of this paper or as a stand-alone perspective where ideas can be discussed in more detail is debatable.

Line by line comments:

Lines 69-77: describes GOLT but lacks statement about likelihood of this hypothesis and same for the MASROS hypothesis, I think a couple of lines/some references showing studies that support or don't support these two hypotheses would be useful here.

Line 113: consider replacing "theories" to "hypotheses"

Line 126-127: Just to clarify the fish in the AG are currently smaller than the GO fish, or projected to get smaller? If the former then I understand the relevance to compare the two population but since its written "estimated to be 14-20% smaller" it wasn't completely clear.

Line 139-140: Is locomotor performance predicted to scale in a similar way to metabolism?

Figure 1: This map shows maximum sea surface temperatures but do you have information about the duration of these maximum temperatures or the variability of temperatures at these two locations, i.e. how often were the fish experiencing these temperatures, if the fish rarely experience these maximum temperatures or for only short durations will they have adaptations to them? Some information/discussion around this would be nice to include.

Lines 166-171: what is the size at maturity for these two species and the maximum sizes? Does the size range you used represent this full range and if not why was this range chosen? Also if the fish from the AG are smaller then by size matching fish then are you excluding the biggest fish in the GO population as the assumption would be they don't exist in the AG?

Lines 175-182: How many fish were housed in each tank?

How many individuals were there within each temperature treatment and were they of equal sizes within treatments?

Line 183: Since fish were collected at different time periods then I assume the experiments were also done at different times, it would be useful to include a timeline (in the supp) to show this. And do you think seasonality would have affected the performance of fish (not only in terms of temperature).

Lines 184-191: I understand that it wasn't possible to collect fish from the GO at 35.5 but would it not have been more comparable to have done the same acclimation on the fish from AG (i.e. collected extra fish at 31.5 and increased temp to 35.5 and held for 3 weeks)?

Lines 292-296: Would be nice to include in the supp a figure of how SMR and MMR were determined. Do you think SMR is reached in the timeframe used in these experiments? 8 hours from being placed in a respirometry chamber and, during daylight hours I don't imagine will give a true SMR. Do you have any supplementary data to support this? Otherwise I would suggest calling it RMR. Some extra details on how the MMR was calculated would also be good to include, e.g. was the whole slope for the entire Ucrit speed used? Or a shorter time interval? Or something else..

Lines 308: "linear mixed effect model"

Fig.2: long term tolerance limit less than 35.5 for *S.ghanam* – suggests there is some acclimation/adaptive change between populations. What is the thermal tolerance (short or long term) for these two species and has it previously been shown to differ between these two populations?

Results: for me all the stats in the text of the results make it hard to read and understand, although perhaps unconventional maybe these can be move to a table/supp to make it clearer?

Figures: would be useful to include sample size in figure legend and for the box plots to have the data points overlaid so its easier to see the distribution of the data. Figure 1 and 2 could also have a description of the dashed grey lines and maybe they aren't needed between the SMR and MMR, unless there for a specific purpose?

Figure 4,5 and 6: I understand the reasoning for pooling the data from the two sites but if you looked at each site independently do the relationships remain the same? For *S.ghanam* in particular since you miss data from the highest temp at one of the sites you increase the variance of the other two temperatures compared to this one, does your statistical test account for this? Also you are aiming to look at the allometry between traits but in two very different populations, one that has the "full" size range and has not undergone intense selection for high temperatures and one with a overall smaller size which has already undergone selection, would you expect these two to scale equally? Some justification or clarification of the reasoning for this would be nice to include.

Line 486: "two adaptive pathways", even though this is an introduction to the results you could state what these two pathways are here and then go into more detail afterwards.

Line 493: you mention a direct connection between these two locations, is there evidence of genetic mixing between your populations or are they genetically distinct?

Line 494: The fact that 12% of species do exist in the AG suggests these species have been able to adapt or acclimate to the warmer temperatures, even if the others haven't (despite your prediction in the introduction). Some discussion around this would be beneficial.

Lines 510-520: The conclusion here that *L.ehrenbergii* doesn't have any adaptive pathways for surviving high temperatures can not be said as the authors clearly show that this fish is still within it optimal temperature range at the highest temperatures tested. If the species was tested at a higher temperature still then maybe differences could be seen between populations (like with the other species tested).

REVIEWER COMMENTS

Reviewer #1 (Remarks to the Author):

This study examines 10 metabolic and physiological performance metrics in two reef fish species living in Gulf of Oman (GO) and Arabian Gulf (AG), aiming to assess whether AG populations, typically exposed to extreme summer temperatures (35C) have developed adaptations to improve their performance. The authors aim to test two oxygen limitation theories (gill oxygen limitation and MASROS – maintain aerobic scope) as a potential explanation for decreasing maximum body sizes of fishes in warmer temperatures. They find limited support for either of the two oxygen limitation theories and instead suggest an alternative theory based on limitation of food availability.

Overall, I found that that the questions, experimental design and execution are generally well performed. It is clear that a lot of work was put into conducting these experiments and the question is generally interesting to the wide audience. This study could become an important paper, but it needs significant work on the presentation of results and discussion.

ANSWER: Thank you for the insightful comments. We agree with the sentiment that this manuscript is an important contribution, which is strengthened by the helpful suggestions. We have answered and address all comments provided and are thankful for your help in improving the manuscript.

My major comments are:

1) Data availability: First and most importantly, I could not find the data available for review, so could not play with the results and analyses and see whether they make sense to me. I could see that the code (in a Word file!) was actually provided in the supplementary material, but without the original data and in this kind of .docx file it is useless. It is essential that for the next round of review the full data and code are made available to reviewers. It is easy to do that by zipping the entire R project and uploading it for review.

ANSWER: Thank you for this suggestion. The double-blind review criteria initially prevented us from releasing the data through a typical data repository which include authorship information. As requested, we have now included the entire R project for review in the supplemental files including annotated code and source data.

2) **2a)** Mixed effect model structure: I am not sure why the authors have used such a convoluted four step method to account for covariation of physiological metrics with body size. Without having the data to assess these steps it was impossible to understand whether this method worked. However, the authors should redo the analyses and include body size in main mixed effect models as a numeric fixed effect. Then you can check for interactions between body size and temperature (i.e. – is scaling different at different temperatures) and regions. Once body size is included in the main analyses you do not need to run separate and confusing analyses regressing raw data against body size. **2b)** Moreover, it is not clear why temperature is treated as a random effect in these additional body size models (random effects should have at least 5 levels of data and you have 2 or 3, depending on the species). As you are citing Wootton et al. study, I suggest looking at their statistical analyses, where physiological measures are tested across temperatures and where body mass is included as a covariate in the models. The data and R code used for the analyses are publicly available. **2c)** Second, I am not sure why the individual ID is included as a random effect in the model. Is this to account for overdispersion or were the repeated measures taken for each individual for a specific metric? **2d)** Was there evidence for overdispersion and was that tested? **2e)** It was not entirely clear to me whether fish were individually tagged and separated for the experiments.

ANSWER:

2a) We apologize for lack of clarify as that appears to be the main issue here. To clarify, we ran two types of comparisons. One comparison (A) which had size fully included in the models (as suggested here) and one Comparison (B) which controlled for putative effects of size.

Comparison (A): For our scaling examinations (i.e. to test GOL and MASROS predictions) we were explicitly interested in the interactive effect of the body-size covariate. This scaling section of our manuscript therefore included body size in the main mixed effect models as a numeric fixed effect. As suggested by the reviewer, we then checked for interactions between body size and temperature and regions (this was also the approach taken by Wootton et al). We then explored the GOL and MASROS predictions for mass scaling relationships by comparing slopes of trendlines using the `lstrends` function on the output of the main fixed effect models (see e.g., Figures 5,6 and Tables S1,S2). The `lstrends` comparison is recommended as it explicitly allows for comparison of scaling slopes as opposed to estimated marginal means (see details in <https://cran.r-project.org/web/packages/emmeans/emmeans.pdf>, and also <https://cran.r-project.org/web/packages/emmeans/vignettes/interactions.html#covariates>).

Comparison (B): For our performance examinations, we needed to facilitate a balanced comparison and understanding of AG versus GO performance across temperatures which was unaffected by the well-documented regional differences in body size (Brandl et al 2020, D'Agostino et al 2021). Thus, in this analysis we were explicitly interested in comparing responses that were conditional on the body-size covariate. Importantly, controlling for (body-size) covariates via baseline covariate adjustment has been shown to increase statistical power in randomized trials and protect against imbalances in covariates (Hernández et al 2004; Kahan et al 2014; Momal et al 2023, Tackney et al 2023, see reference details in reference list below). This can be done either by i) including the conditional covariate as an added factor in the model, OR by ii) performing residual regression analyses to eliminate the conditional effect of the covariate (Reist 1985; Hernández et al 2004; Tsiatis et al 2008; Tackney et al 2023). The latter approach (ii) increases statistical power by reducing the number of model factors, by reducing model complexity, and by increasing available degrees of freedom. For that reason, we used residual regressions to control for body size while also increasing statistical power. Our approach to control for body size follows standard protocols for the past 40+ years as described by e.g., Reist 1985 (see reference details below). However, we acknowledge that our rationale was not adequately explained in the manuscript and that it was not easy for reviewers to evaluate the appropriateness and effectiveness of our scaling approach, and for this we apologize. **To ensure full transparency, we have therefore followed the suggested example of Wootton et al 2022 and opted to automate our analyses in R. We now provide the annotated r-script and raw data as supplemental data files. Readers of the manuscript can now readily examine our covariate approach using our step-by-step script and data, which includes generation of graphs for visual evaluation.** Additionally, we have included a rationale for our statistical approach in the methods section, line 292, which read: “.. for the 10 examined performance metrics there is currently no consensus for the relationship between size and temperature^{47,53,96}. Thus, in order to avoid dictating the combined effect of body size and temperature and to facilitate a balanced comparison of AG versus GO performance within and across temperatures that were unaffected by regional size differences, data were size adjusted in R using four automated steps (see r-script and raw data in supplemental material)... As we were explicitly interested in comparing responses that were conditional on the body-size covariate, we opted to use this baseline covariate adjustment approach to statistically remove its effects from our models⁹⁷, as this approach has been shown to control against imbalances in covariates and increase statistical power in randomized trials by reducing the number of model factors, by reducing model complexity, and by increasing available degrees of freedom⁹⁷⁻¹⁰²”.

Importantly, we also want to acknowledge that the former statistical approach (i) suggested by the reviewer is equally suitable. **For that reason, we have now included a second supplemental R-scripts with data which follow the reviewer suggestion of adding body size as a fixed effect covariate. The**

results of the new analyses closely match our current interpretation but shows general reductions in model fit due to reduced statistical power. For instance, metabolic model fits are reduced from marginal $r^2 = 0.89-96$ to marginal $r^2 = 0.60-63$, while model fits on swimming data are changed from marginal $r^2 = 0.65-67$ to marginal $r^2 = 0.46-68$ across the two species. Both analytical approaches result in the same conclusions, highlighting the suitability of both approaches in this case. Given increased model complexity and reduced statistical power, we would prefer to keep the strongest analyses (ii) in the main manuscript and the alternative (i) in the supplemental materials, but would be open to the reverse if so requested.

2b) Thank you for highlighting the potential issue of using random effects with only three levels of data. We agree that more than three levels are prudent. In the single-metric scaling analyses, each individual fish is only represented once and linear models with a random effect assigned to a parameter with only one measurement can't distinguish between the random effect and the residual error. As a result, these models require population, temperature, or the combination of both to be treated as random. To ensure adequate levels of data we have therefore updated these models to include both population and temperature in the random effects, thereby providing ≥ 5 levels of data as required. This update has been included in our analyses section (line 341) and our new data tables (Table S1 and S2) and caused no change in significance of our results.

2c) We apologize for the lack of clarity here. In the multi-metric performance analyses, individual ID was included to account for multiple datapoints stemming from the same individual. We have now clarified this in line 337 stating "Individual ID was included as a random effect to account for multiple datapoints stemming from the same individual"

2d) Evidence of overdispersion was tested as the ratio of residual variance to degrees of freedom from the summary model output. Values substantially above 1 are indicative of overdispersion and all our models had values ≤ 1 except for one single case: The *S. ghanam* metabolism model initially had a value > 20 , which reduced to 0.98 after a Box-Cox transformation was applied. There was no evidence of overdispersion in any of our final models and this approach has now been clarified in our analyses section (See line 358): "Evidence of overdispersion was tested as the ratio of residual deviance to degrees of freedom from the summary model outputs, with values above 1 indicative of overdispersion. Data that did not initially comply with assumptions were Box-Cox transformed and ...All final model data met assumptions"

2e) These fish were housed in groups of four individuals and tagged with visible elastomer tags, thereby allowing individual identification. This has now been clarified in the manuscript line 185: "Fish were housed in groups of four fish per tank and tagged with visible elastomer tags (Northwest Marine Technology, Inc.) to allow individual identification"

3) Use of 35.5C temperature without long-term acclimation. The GO individuals do not experience 35.5C temperatures. In this study they were acclimated to such temperatures for 21 days, but this is absolutely not enough for the question of long-term acclimation and adaptation. The fact that one species showed high mortality (100% mortality?) just confirms that. Again, as you refer to Wootton et al. study, they showed that 3 generations of acclimation may be needed before SMR is fully acclimated to new temperatures. I understand that 3 generations may not be feasible for many studies, but we should not interpret short-term acclimation studies in the long-term adaptation context then. I strongly suggest focusing main results on 27 and 31.5C temperatures and including 35.5 just as additional special case for the AG species only (because they are naturally acclimated to these conditions).

ANSWER: Thank you. It is important to note that this study did not attempt to long-term acclimate or adapt GO fishes to 35.5C. Rather, this study examined the relative capacity of AG and GO fishes to function in 35.5C water as an indication of thermal adaptation which have occurred explicitly in the AG population (not the GO population). The three-week acclimation period used here was purposely chosen because previous research has demonstrated that if a tropical reef fish species is capable of seasonal adjustments to the cardiorespiratory pathways (i.e. SMR, MMR), then this will usually occur within a three-week period (see details in Johansen et al 2021, reference provided below). This notion is confirmed by the fact that AG and GO *L. ehrenbergii* had equal performance at 35.5C after the acclimation period. For *S. ghanam*, the fact that AG individuals were able to function under 35.5C conditions, whereas GO individuals could not, provides strong evidence of the adjustments that have occurred explicitly within the AG. We acknowledge that this important distinction was not abundantly clear and have therefore made several changes throughout the manuscript. For instance, our primary hypothesis in the introductions (line 139) now reads: “we aim to elucidate 1) whether reef fishes in the AG have managed to improve metabolic and swimming performance metrics to support survival at elevated temperatures”. Within the methods, line 194 now reads: “In order to compare the relative capacity of AG and GO fishes to function in 35.5°C water as an indication of thermal adaptation which have occurred explicitly in the AG population, GO fishes were collected during peak summer at $31.5 \pm 0.5^\circ\text{C}$ in GO and acclimated for three weeks to $35.5 \pm 0.5^\circ\text{C}$ to maximize acclimation potential (following Johansen et al., ⁷⁷). The three-week acclimation period was chosen because previous research has demonstrated that maximal adjustments of metabolic metrics will usually occur within a three-week period in tropical reef fish (sensu Johansen et al., ⁷⁷)”. Within the discussion, line 515 now reads: “our findings suggest that two divergent pathways have facilitated fitness and survival of fishes in the AG. Explicitly, *L. ehrenbergii* showed no population level signs of adaptation to life at the elevated temperatures found within the AG, instead appearing to thrive in the AG due to a high intrinsic species-specific tolerance to temperatures reaching 35.5°C. Indeed, concurrent with increasing metabolic demand for activity (i.e. cost of swimming) caused by rising temperatures ¹¹⁷⁻¹¹⁹, this species showed an analogous increase in surplus energy available to fuel ecological functions both in AG individuals collected at 35.5°C and in and GO individuals after short-term acclimation to 35.5°C.... As a result, there was no evidence of oxygen limitation in this species and no change in any swimming performance metric examined (neither optimal, burst or critical swimming speeds or kinematics), highlighting that the swimming dependent activities needed by this species for survival at +3°C (including foraging and predator evasion) were well within its existing performance window”. Line 528: “Contrary to *L. ehrenbergii*, *S. ghanam* showed clear signs of a population level response to survival at the elevated temperatures of the AG both in terms of oxygen supply capacity and swimming performance. While this species also showed increasing metabolic demand for swimming with rising temperature, only the AG individuals were able to maintain AS up to 35.5°C. This metabolic surplus allowed the AG individual to fuel critical swimming activities at 35.5°C, whereas the GO individuals demonstrated a complete inability to survive at the same temperature”. We have also added the following caveat to our discussion section line 594: “..although the fact that only the AG individuals could be collected and directly tested at 35.5°C renders some uncertainty for the mass scaling exponent at this temperature”.

4) Presentation of results. To be honest, current presentation of results is too hard to read and comprehend. It mechanically reports various statistics, which should be instead placed in a table. Each section is repetitive and the reader quickly loses track of what is found and why. Since you present quite a lot of different results, it is essential that results are presented clearly, visual summaries are informative and repetitive statistics are reported in a table. In addition to or instead of performing multiple planned comparisons, please include body weight in the main model and use mixed effect model summaries and

plots of highlight statistical model outputs in a more intuitive way. You can use packages and functions “effects”, “predict” and show model predicted responses with uncertainty, with raw data in the background. This way you can show variation in the actual raw data and various statistics predicted by the model (you can use the publicly available code from Wootton et al for the plots, if it is useful). Please move all the statistical reporting from the Results text into tables and figures, and instead present results in plain language, indicating main points, your hypotheses and main findings.

ANSWER: Thank you for these suggestions. We acknowledge that this manuscript presents a lot of complex results, which can be difficult to follow. The statistical suggestions for clarity have been addressed above. Additionally, we have tried different ways of presenting the data more intuitively and our initial presentation of results was in-fact already a slimmed down version of the full results (provided in the original supplemental data). Based on the excellent suggestions provided here (and mirrored by the second reviewer below), we have now opted to move nearly all statistical reporting into tables and figures and instead present results in simplified language focusing on our main findings in relation to our aims and hypotheses. As a result, the new presentation of results is more intuitive and easier to follow. For those readers who prefer the more technical text with statistical reporting, we have also kept the full technical results section as a supplemental file.

5) **5a)** Introduction of a new hypothesis for temperature effect on maximum body size. Your study has great results about various physiological metrics across temperatures, but you did not measure resource availability and usage. For me your main result is that SMR across 4.5C temperature difference (ie between 27 and 31.5C) is generally the same, which is an important and interesting finding. **5b)** But it does not fit your subsequent speculation about resource limitation driven by temperature. There is no data in this study to support it and it is based on wrong information. For example, you do not present data on maximum body sizes in this study. Also to justify resource limitation ideas, you assume Q10 of 2-3 and 23-39% increase in basal energy demand across 3C temperature difference (line 647). But your own data shows that for SMR (i.e. basal energy needs) Q10 is 1! There was no difference across 4.5C in temperature. So why should basal energy demand increase? **5c)** Also, two theories on oxygen limitation (line 69) are not the only theories aiming to explain why body sizes are smaller in warmer temperatures. The main competing theory is life-history based explanation, again presented in Wootton et al., but not mentioned in your study. It suggests that faster juvenile growth and earlier maturation diverts resources to reproduction instead of growth (either for adaptive or some limiting reasons, which is a separate and complicated question). Moreover, there are already multiple hypotheses suggesting resource limitation as an explanation for smaller body sizes in warm temperature. See Verberk et al. 2021 and Audzijonyte et al. 2019 for reviews on these theories. Therefore, I strongly believe that the speculative part of “resource limitation” and “consolidating theory” should be completely removed from this paper. It fits better for a separate paper, if the authors wish to pursue it. **5d)** Focus on your interesting results and interpretations, such as very little difference in performance rates across 4.5C temperatures (assuming this result remains unchanged once body mass is included as a covariate in the main model), different degrees of adaptation in the two species, better discussion on swimming performance metrics. Please make sure you summarise your key findings in the discussion again and discuss them specifically, comparing your findings with other studies. Currently, the discussion is too general and not relevant to your findings.

ANSWER:

5a) Thank you for this suggestion. We are happy that the reviewer sees the importance of our results and we agree that the lower-than-expected increase in basal metabolic rates are intriguing. We have therefore followed the reviewer suggestion by incorporating this interesting finding in our abstract, results, and summary discussion paragraph. We have also added an entirely new paragraph to the

discussion. The abstract now reads: “Comparing *Lutjanus ehrenbergii* and *Scolopsis ghanam* from AG to individuals from typical present-day coral reef temperatures in the Gulf of Oman (summer max 32.0°C) across 27.0, 31.5°C and 35.5°C, our data...show a lower-than-expected rise in basal metabolic demands and a right-shifted thermal window facilitating a capacity to maintain oxygen supply and aerobic performance to 35.5°C”. Furthermore, our summary discussion paragraphs (line 470) now reads: “Using the worlds warmest coral reefs in the Arabian Gulf (AG) as a natural laboratory for ocean warming, our data reveals two adaptive pathways across 10 metabolic and swimming performance metrics that appear to have facilitated survival at elevated temperatures, including a lower-than-expected rise in basal metabolic demands...” Similarly, line 497 reads: “Despite predictions, neither species showed substantial increases in basal metabolic demands (SMR) and a lower-than-expected temperature sensitivity quotient (Q10) of 1.33 to 2.04 across the range of temperatures examined here. Given the ectothermic nature of most fishes and direct correlations between temperature and biochemical reaction rates⁴¹, there is an expectation of exponentially rising metabolic demands as a consequence of ocean warming. However, it is also well known that most organisms exhibit phenotypic plasticity in the expression of metabolism in response to environmental challenges, which may allow suppressed post-acclimation basal demands across temperature within the constraints of stoichiometry in relation to body size^{41,48,117,121}. As such, limited increases in SMR and energetic demand may promote survival in high temperature habitats and although direct empirical evidence is limited, suppressed SMR has been shown in laboratory zebrafish acclimated to elevated temperatures over six generations (see Wootton et al¹). Intriguingly, based on evaluations of thermal sensitivity in hundreds of fish species across latitudes, the average post-acclimatization Q10 equates to ~1.65-1.91^{41,48,121,122}, Killen et al 2010, Clarke & Johnston 1999). Our results closely resemble these post-acclimatization values with a mean Q10 of 1.71, supporting the notion that our fishes were indeed fully acclimated to the temperatures ranges examined and that reductions in basal metabolic demands may aid survival of reef fishes as temperatures rise”.

5b) We respectfully disagree that our data does not support resource limitation as a driver of size reductions. As described above our data shows a mean Q10 of ~1.71 (not 1.0 as suggested) for these fish and existing literature show post-acclimatization Q10s of 1.7 to 1.9 in hundreds of arthropods, mulloscs, fish, amphibians and reptiles across latitudes (see Brown et al 2014, Clark & Johnson 1999; Killen et al 2010; White et al 2006), highlighting an increased need for energy in warmer waters. Additionally, since submission of this manuscript, two important papers have been published with similar supporting evidence for resource limitation as a driving factor for size reductions: Queiros et al (2024) found that “food resources and temperature are major environmental drivers that can dramatically increase energy expenditures of fishes and disturb their energy balance in a scenario of future climate change” and state that “the results also provide experimental evidence that such challenges to energy balance may contribute to the ongoing shrinking of fish populations”. Likewise Skeeles & Clark (2023) found that “growth improved marginally with additional energy.. thereby providing evidence for a role for energy reallocation in the deceleration of adult growth. Interestingly, additional dietary energy had a disproportionately larger effect on the growth of fish that matured at a greater size, revealing size-dependent variance in energy acquisition and/or allocation budgets at summer temperatures”. As described in our study, these increases in energetic demand and food resource needs can be fully offset by the size reductions observed in the AG, and resource limitation is therefore arguably a strong explanation for observed patterns. We have now implemented these new publications as supportive evidence of our consolidating energy acquisition theory several places in the discussion, including line 615: “For instance, predatory fishes which typically deploy a feast and famine approach to foraging¹²⁷ may be forced to reduce meal sizes to protect aerobic scope due to the exponential costs associated with digestion of large prey^{34,52,128-130} and may be unable to significantly increase capture rates of (smaller) prey¹⁰⁷,

thereby placing a natural upper limit to energy acquisition. Indeed, recent empirical work on sardines have shown 2-3-fold increases in energy expenditure needed to capture smaller prey and suggested that this bottom-up effect may help explain patterns of shrinking in wild fishes (Queiros et al 2024). Similarly, continuous grazers such as herbivorous reef fishes typically spend the majority of their waking time foraging under current day temperatures (some more than 80%¹³¹⁻¹³³), likely hindering a further 20-40% increase in foraging effort and energy acquisition to match warming-induced metabolic demands”.

Importantly, we do acknowledge the reviewer’s viewpoint that temperature driven increases in energetic demand will likely be lower than a factor of 2-3, and arguably closer to a Q10 of 1.6-1.9. Thank you for pointing this out. As such, we have updated our model calculations to reflect this important viewpoint in line 666: “Using established equations for temperature sensitivity of fish metabolism a 3.0°C increase in ocean temperature is expected to incur a 16-30% increase in basal energy demand based on a Q10 of 1.65-2.40^{41,48,123-125}. Simultaneously, most fishes are thought to have a metabolic mass-scaling exponent of ~0.8-0.9 (e.g. $\ln(\text{smr}) = 0.8 * \ln(\text{mass}) - 5.43$ ⁴⁸), and our data confirms those expectations with an average exponent of 0.9 ± 0.05 (\pm S.E.M., see Fig. 5). As a result, any given species could offset the expected increase in absolute energy demand by reducing mass by ~18-28%. Accordingly, AG cryptobenthic fishes are known to be 9.1-40.2% smaller in mass than GO conspecific fishes⁷¹ and larger bodied species such as *L. ehrenbergii* and *Pomacanthus maculosus* are known to be ~32-46% smaller in AG (based on published TL reductions of maximum body size and weight-length relationships^{77,166,167}). Consequently, the observed size reductions in AG fishes, which match the 14-40% size reductions projected globally, theoretically all but eliminates the need to increase absolute energy acquisition above the evolved upper limit for prey capture and digestion for any given species (Figure 7)”.

5c) This is first study to validate cardiorespiratory theories for size reductions against the 6000-year history of the hottest tropical coral reef on Earth. We then use complimentary existing knowledge to try explain what might be driving observed patterns. Importantly, although we do address multiple other theories, including “ecological metabolic theory”, “resource limitation theory” and “temperature-dependent resource allocation”, in relation to our findings, we do not attempt to test all theories ever produced such as those explicitly related to e.g., reproduction, as that would be beyond the scope of a single paper. We also do not attempt to prove, conclusively, whether our consolidated resource acquisition theory is 100% universal. We merely provide a theoretical basis and stepping stone for new research in order to help move the needle forward on this critically important question. As an imminent and urgent climate-change issue this topic demands expedient scientific scrutiny, and removal of our theory would effectively shield it from discussion and examination for the next many years until these authors are able to make headway themselves. Likewise, Reviewer 2 also see merit in our “compelling theory”. As such, we respectfully disagree with the suggestion of removing all notions of a consolidating energy acquisition theory from this paper. Incidentally, the reviewer mentions the life-history theory that faster juvenile growth and earlier maturation may divert resources to reproduction instead of growth for “some limiting reasons”, and our consolidating energy acquisition theory actually provides a basis for testable hypotheses and predictions that would explicitly look at such “limiting reasons”. For instance, fishes exploiting the same, easily digestible prey sizes and types across adult sizes are hypothesized to experience limits on energy acquisition from maximal sustained per-capita prey capture rates, thereby causing a limitation in the total energy available to allocate towards reproduction and growth (see hypothesis 1, Box 1). Thus, to avoid stifling scientific discussion and progress we therefore maintain that this theory is best be presented to the wider scientific community for scrutiny.

PS: in order to better portray existing hypotheses in relation to our energy acquisition theory, we have now rephrased and more clearly emphasized mentions of e.g. resource-limitation theory, temperature-

dependent resource allocation theory and metabolic theory throughout the discussion. We have also added and incorporated several new references, including:

- Lee, K. P., Jang, T., Ravzanaadii, N., & Rho, M. S. (2015). Macronutrient balance modulates the temperature-size rule in an ectotherm. *The American Naturalist*, 186(2), 212-222.
- Skeeles, M. R., Scheuffele, H., & Clark, T. D. (2023). Supplemental oxygen does not improve growth but can enhance reproductive capacity of fish. *Proceedings of the Royal Society B*, 290(2010), 20231779. Doi: 10.1098/rspb.2023.1779
- Skeeles, M. R., & Clark, T. D. (2023). Evidence for energy reallocation, not oxygen limitation, driving the deceleration in growth of adult fish. *Journal of Experimental Biology*, 226(13). Doi: 10.1242/jeb.246012
- Queiros Q, McKenzie DJ, Dutto G, Killen S, Saraux C, Schull Q (2024). Fish shrinking, energy balance and climate change, *Science of The Total Environment*, 906, 167310. Doi: 10.1016/j.scitotenv.2023.167310

5d) Thank you for this suggestion. We agree that additional discussions on performance rates across temperature might be useful and we have now included these topics in the discussion summary and as individual sections within the discussion. For instance, line 471 reads: “..our data reveals two adaptive pathways across 10 metabolic and swimming performance metrics that appear to have facilitated survival at elevated temperatures, including a lower-than-expected rise in basal metabolic demands and a right-shifted thermal window facilitating a capacity to maintain aerobic scope and critical swimming speeds into higher temperatures”. Additionally, line 518 now reads: “concurrent with increasing metabolic demand for activity (i.e. cost of swimming) caused by rising temperatures¹¹⁹⁻¹²¹, this species showed an analogous increase in surplus energy available to fuel ecological functions both in AG individuals collected at 35.5°C and in and GO individuals after short-term acclimation to 35.5°C. Specifically, as cost-of-transport (COT) and cost of critical swimming (MMR at U_{crit}) increased across temperatures, so too did the aerobic scope (AS). As a result, there was no .. change in any swimming performance metric examined (neither optimal, burst, or critical swimming speeds or kinematics), highlighting that the swimming dependent activities needed by this species for survival at +3°C (including foraging and predator evasion) were well within its existing performance window”. Discussions of swimming are also included in lines 528: “Contrary to *L. ehrenbergii*, *S. ghanam* showed clear signs of a population level response to survival at the elevated temperatures of the AG both in terms of oxygen supply capacity and swimming performance. While this species also showed increasing metabolic demand for swimming with rising temperature, only the AG individuals were able to maintain AS up to 35.5°C. This metabolic surplus allowed the AG individuals to fuel critical swimming activities at 35.5°C, whereas the GO individuals demonstrated a complete inability to survive at the same temperature. Interestingly, the GO individuals also suffered >20.0% reductions in AS and concurrent 17-25% reductions in all swimming performance metrics at 31.5°C, suggesting that the ecology of this species allow individuals to survive prolonged periods of reduced swimming performance during summer. Indeed, previous work has shown that some fishes in the AG region broaden their diet in the spring but then narrow their diets in the summer¹²³, while others upregulate foraging and energy storage in the spring, but then downregulate costly swimming activities during summer¹²⁴. This latter trait was mirrored in the AG *S. ghanam* as there were no differences in metabolic and swimming performance of the GO individuals at 31.5°C and the AG individuals at 35.5°C, suggesting that population level responses to elevated temperatures have led to a significant right-shift in the thermal performance window of this species¹²⁰, but no other changes in the ecological performance or survival strategy during peak summer”.

6) Comparison of GOLT and MASROS hypotheses. GOLT has been misinterpreted and misunderstood by many researchers. I am not a big fan of it myself, but it is important to interpret it correctly. It is not very different from MASROS, because it suggests that maturation is triggered by decreasing oxygen availability in the blood (due to different scaling of gill surface area and body volume) and further growth is reduced. It does not say that large fish do not have

aerobic capacity left, because obviously that would be un-adaptive and a silly thing to do. So both hypotheses are talking about the overall maintenance of aerobic scope and I am not sure why they are treated differently.

ANSWER: We apologize for the lack of clarity here and agree that accuracy is paramount. The sentence was meant to refer to aerobic surplus for growth and we have therefore rephrased the sentence from “*aerobic surplus approaches zero*” to “*aerobic surplus becomes inadequate to support further growth*”. Although the GOLT and MASROS theories are similar in some ways (e.g. they both relate to oxygen delivery to muscles and tissues as described in our manuscript and others, see e.g. Skeeles et al 2023), they are still two distinct hypotheses with distinct meanings and explanations for size reductions in fishes (see Verberk et al 2021 Figure 3 for explicit graphical depictions of how these models and hypotheses differ). Statements in Verberk et al 2021 include “*in the Pauley (GOLT) model “constraints on growing to a larger size are considered to be insurmountable, arising from geometric constraints on gill surface area scaling, and growth ceases when maintenance metabolism converges to supply capacity”*”. “*In the MASROS model, animals still have aerobic scope left when reaching maximum size, which is considered to be a safety margin when animals face demanding but transient conditions (e.g. disease, episodes of hypoxia, predator attack, and possibly part of reproduction)*”. As such it is clear that there are critical differences between these two related hypotheses. Given the breadth of high-profile, high-impact studies that have discussed the pros and cons of these two hypotheses and also the differing steps needed to test each (which laid the foundation of this study), these two models arguably differ (and are treated as such by the wider scientific community). We refer to Verberken et al 2021 for a good overview of how they differ within the context of our study.

Minor comments:

- It is not really clear that the maximum size of many species is reducing due to temperature. The study you cite in support of this (line 107) is Cheung et al. 2013 and it does not use any empirical data for their claims.

ANSWER: Apologies for the lack of clarity in this sentence. Although it is worth noting that the Cheung 2013 modeling paper is indeed firmly based on empirical data (please see their supplemental data section for details), we have opted to rephrase this sentence and added several additional references to bolster support, including the most recent IPCC report. The sentence in line 110 now reads: “*There is little doubt that species are already responding to changing environmental conditions including rising temperatures⁵⁹ and that the maximal body size of many species appears to be reducing^{17-20,22,23,60-63}*”.

- On line 470 you conclude that *S. ghanam* was partially limited by mass, but it is not clear why this conclusion has been reached, as COT increased with mass at all temperatures. It is essential not to add ad hoc interpretations to our results. Therefore please include mass in the main model and test for interactions between mass and temperature and region. This way you can discuss whether there was any significant interaction. It is still not clear what such interaction would actually mean in terms of limitations. In order to show that e.g. fishes oxygen availability become more limited in larger sizes we would need to show that the ABILITY to uptake oxygen does not scale as fast as the demand for oxygen. I am not sure whether anyone has done that.

ANSWER: We apologize for the lack of clarity here. This is not an ad hoc interpretation, but a general statement of results as it pertains to our predictions, which we have now rephrased for clarity. Line 428 reads: “*We predicted that larger fish should experience greater reductions in metabolic performance (following the GOL hypothesis) and swimming performance (following the MASROS hypothesis) with*

each degree increase in temperature”, but results show (line 430) that “contrary to predictions, *L. ehrenbergii* was not limited by mass at elevated temperatures and there were no recorded negative consequences for larger individuals”, whereas (line 446) “Contrary to GOL predictions, *S. ghanam* was also unaffected by mass in metabolic performance but did follow MASROS predictions of being limited by mass in swimming performance”. We do acknowledge that this may not have been abundantly clear from the original reporting, partly due to the complex nature of our results (see other reviewer comments on the presentation of results). In order to increase clarity of our entire results section we have therefore followed reviewer suggestions of moving all the statistical reporting from the text into tables and figures, and instead presenting results in plain language, indicating main points, hypotheses and main findings. Please see the new Table S1-2 for model outputs. We believe the new presentation is clearer and easier to comprehend. However, as this approach does not convey detailed results with exact accompanying statistics as many readers are accustomed, we have opted to also provide an updated version of the detailed written results section in our supplemental documentation. This way, the reader can now choose between a plain language results section in the main manuscript, or an accompanying in-depth detailed results section in the supplemental material.

- On line 561 – Wootton et al. did not say that existing GOLT evidence comes from short term studies, they did not discuss existing evidence for GOLT at all. They said that increased energetic or oxygen demands in warmer water are shown mostly in short term studies, yet when they allowed for multi-generational acclimation, they found that there is no difference in SMR between 4C temperature. This is also largely your finding, yet for some reason you still assume that higher temperatures will increase energetic demands.

ANSWER: Thank you for finding this error. This was meant to say Audzijonyte et al. not Wootton et al. This has now been corrected. We also appreciate the notion that Wootton et al found no significant change in SMR across 4 temperatures, which is a very interesting finding and worthy of further exploration. As stated above we have now included this finding in our discussions with an entirely new section.

References:

- Binning, S. A., Ros, A. F., Nusbaumer, D., & Roche, D. G. (2015). Physiological plasticity to water flow habitat in the damselfish, *Acanthochromis polyacanthus*: linking phenotype to performance. *PLoS One*, *10*(3), e0121983.
- Brown, J.H., Gillooly, J.F., Allen, A.P., Savage, V.M. & West, G.B. (2004). Toward a metabolic theory of ecology. *Ecology*, *85*, 1771–1789.
- Brandl SJ, Johansen JL, Casey JM, Tornabene L, Morais RA, Burt JA (2020) Extreme environmental conditions reduce coral reef fish biodiversity and productivity. *Nature Communications* *11*:3832
- Clark, T. D., Sandblom, E., & Jutfelt, F. (2013) Aerobic scope measurements of fishes in an era of climate change: respirometry, relevance and recommendations. *Journal of Experimental Biology*, *216*(15), 2771-2782. Doi:10.1242/jeb.084251
- Clarke, A., & Johnston, N. M. (1999). Scaling of metabolic rate with body mass and temperature in teleost fish. *Journal of animal ecology*, *68*(5), 893-905.
- D’Agostino, D., Burt, J. A., Santinelli, V., Vaughan, G. O., Fowler, A. M., Reader, T., ... & Feary, D. A. (2021). Growth impacts in a changing ocean: insights from two coral reef fishes in an extreme environment. *Coral Reefs*, *40*(2), 433-446. Doi: 10.1007/s00338-021-02061-6

- Grandcourt, E., Al Abdessalaam, T. Z., Francis, F., & Al Shamsi, A. (2011) Demographic parameters and status assessments of *Lutjanus ehrenbergii*, *Lethrinus lentjan*, *Plectorhinchus sordidus* and *Rhabdosargus sarba* in the southern Arabian Gulf. *Journal of Applied Ichthyology*, 27(5), 1203-1211.
- Griffiths, MH, Christopher M Wade, Daniele D'Agostino, Michael L Berumen, John A Burt, Joseph D DiBattista, David A Feary, Phylogeography of a commercially important reef fish, *Lutjanus ehrenbergii*, from the coastal waters of the Arabian Peninsula, *Biological Journal of the Linnean Society*, 2024;, blad170, <https://doi.org/10.1093/biolinnean/blad170>
- Hernández, Adrián V., Ewout W. Steyerberg, J. Dik F. Habbema (2004) Covariate adjustment in randomized controlled trials with dichotomous outcomes increases statistical power and reduces sample size requirements. *Journal of clinical epidemiology*, 57.5, 454-460.
- Johansen, J. L., & Jones, G. P. (2011). Increasing ocean temperature reduces the metabolic performance and swimming ability of coral reef damselfishes. *Global Change Biology*, 17(9), 2971-2979.
- Johansen, J. L., Nadler, L. E., Habary, A., Bowden, A. J., & Rummer, J. (2021). Thermal acclimation of tropical coral reef fishes to global heat waves. *Elife*, 10, e59162. <https://doi.org/10.7554/eLife.59162>
- Kahan, B.C., Jairath, V., Doré, C.J. *et al.* (2014) The risks and rewards of covariate adjustment in randomized trials: an assessment of 12 outcomes from 8 studies. *Trials*, 15, 139. <https://doi.org/10.1186/1745-6215-15-139>
- Ketabi, R., & Jamili, S. (2017) *Lutjanus ehrenbergii* (Peters, 1869). <http://hdl.handle.net/1834/36055>
- Killen, S. S., Atkinson, D., & Glazier, D. S. (2010). The intraspecific scaling of metabolic rate with body mass in fishes depends on lifestyle and temperature. *Ecology letters*, 13(2), 184-193.
- Momal R, Li H, Trichelair P, Blum MGB, Balazard F. (2023) More efficient and inclusive time-to-event trials with covariate adjustment: a simulation study. *Trials*, 24(1), 380. doi: 10.1186/s13063-023-07375-0. PMID: 37280655; PMCID: PMC10245605.
- Nzioka, R. M. (1983). Some biological aspects of (*Scolopsis ghanam* [Forsk.] perciformes: Scolopsidae). *East African Agricultural and Forestry Journal*, 48(1-4). Doi: 10.1080/00128325.1982.11663104
- Queiros, Q., McKenzie, D. J., Dutto, G., Killen, S., Saraux, C., & Schull, Q. (2024). Fish shrinking, energy balance and climate change. *Science of the Total Environment*, 906, 167310.
- Reist, J. D. (1985). An empirical evaluation of several univariate methods that adjust for size variation in morphometric data. *Canadian Journal of Zoology*, 63(6), 1429-1439
- Roche, D. G., Binning, S. A., Bosiger, Y., Johansen, J. L., & Rummer, J. L. (2013). Finding the best estimates of metabolic rates in a coral reef fish. *Journal of Experimental Biology*, 216(11), 2103-2110.
- Rummer, J. L., Binning, S. A., Roche, D. G., & Johansen, J. L. (2016). Methods matter: considering locomotory mode and respirometry technique when estimating metabolic rates of fishes. *Conservation Physiology*, 4(1), cow008.
- Rummer, J. L., Couturier, C. S., Stecyk, J. A., Gardiner, N. M., Kinch, J. P., Nilsson, G. E., & Munday, P. L. (2014). Life on the edge: thermal optima for aerobic scope of equatorial reef fishes are close to current day temperatures. *Global change biology*, 20(4), 1055-1066.
- Russell, B.C. (1990) FAO species catalogue. Vol. 12. Nemipterid Fishes of the World. (Threadfin breems, Whiptail breems, Monocle breems, Dwarf monocle breems, and Coral breems). Family Nemipteridae. An Annotated and Illustrated Catalogue of Nemipterid Species known to Date. FAO Fisheries Synopsis. No. 125, Volume 12. Rome, FAO. 149 p., VIII plates. <https://www.fao.org/3/t0416e/t0416e.pdf>

- Skeeles, M. R., & Clark, T. D. (2023). Evidence for energy reallocation, not oxygen limitation, driving the deceleration in growth of adult fish. *Journal of Experimental Biology*, 226(13).
- Skeeles, M. R., Scheuffele, H., & Clark, T. D. (2023) Supplemental oxygen does not improve growth but can enhance reproductive capacity of fish. *Proceedings of the Royal Society B*, 290, 20231779. <https://doi.org/10.1098/rspb.2023.1779>
- Steffensen, J. F. (1989) Some errors in respirometry of aquatic breathers: how to avoid and correct for them. *Fish Physiol Biochem*, 6(1), 49-59. Doi: 10.1007/BF02995809
- Steffensen, J. F. (1989). Some errors in respirometry of aquatic breathers: how to avoid and correct for them. *Fish Physiol Biochem*, 6(1), 49-59.
- Steffensen, J. F., Johansen, K., & Bushnell, P. G. (1984). An automated swimming respirometer. *Comparative biochemistry and physiology--Part A: Physiology*, 79(3), 437-440.
- Svendsen, M. B. S., Bushnell, P. G., & Steffensen, J. F. (2016) Design and setup of intermittent-flow respirometry system for aquatic organisms. *Journal of fish biology*, 88(1), 26-50. Doi: 10.1111/jfb.12797
- Svendsen, M. B. S., Bushnell, P. G., & Steffensen, J. F. (2016). Design and setup of intermittent-flow respirometry system for aquatic organisms. *Journal of fish biology*, 88(1), 26-50.
- Tackney, M.S., Morris, T., White, I. *et al.* (2023) A comparison of covariate adjustment approaches under model misspecification in individually randomized trials. *Trials*, 24, 14. <https://doi.org/10.1186/s13063-022-06967-6>
- Tsiatis, Anastasios A., et al. (2008) Covariate adjustment for two-sample treatment comparisons in randomized clinical trials: a principled yet flexible approach. *Statistics in medicine*, 27.23, 4658-4677.
- White, C. R., Phillips, N. F., & Seymour, R. S. (2006). The scaling and temperature dependence of vertebrate metabolism. *Biology letters*, 2(1), 125-127.

REVIEWER COMMENTS

Reviewer #2 (Remarks to the Author):

The system the authors have used in this study is unique and perfect for empirically testing theoretical hypotheses regarding shrinking in fish with increasing temperatures. Overall, I found the paper interesting and adds much needed data to this research area. I do however have some concerns regarding the experimental design and interpretation of the results which could use some clarification (see detailed comments below).

ANSWER: Thank you so much for the thorough and astute comments. We are happy that the reviewer sees the same importance and merit in this study as we do, and have provided useful ideas to help improve the study further. We have answered and addressed all comments provided below and believe the manuscript is greatly improved as a result. Thank you.

The PASLED hypothesis the authors present at the end of the discussion is a compelling one with clearly defined and easily testable predictions, it has foundations in various previous hypotheses but uses the ideas in a new way to predict how reductions in body size will vary depending on the species foraging mode and prey type. Whether or not it should be included in the discussion of this paper or as a stand-alone perspective where ideas can be discussed in more detail is debatable.

ANSWER: Thank you for this comment also. We greatly appreciate the acknowledgement that our PASLED energy acquisition theory is well founded in previous hypotheses, well defined, and that it provides new and easily testable hypotheses. As describe in detail above (see answer to question 5b) there are numerous reasons to include this theory here, including the fact that the authors of this manuscript are not likely to be able to perform all of the suggested tests themselves in the timely manner warranted by the urgency of climate change and ocean warming worldwide. Importantly, we also believe our inclusion of the PASLED hypothesis in this manuscript will spark great discussion and interest from the scientific community, and thus be a benefit to the community at large. We are confident *Nature Communications* is the right venue to share this hypothesis, to act as a discussion base and springboard to move the field forward.

Line by line comments:

Lines 69-77: describes GOLT but lacks statement about likelihood of this hypothesis and same for the MASROS hypothesis, I think a couple of lines/some references showing studies that support or don't support these two hypotheses would be useful here.

ANSWER: Thank you for this excellent suggestion. We have now added a new sentence to line 87 which state: “Despite significant global attention given to both hypotheses, there is currently no consensus as to the capacity of either hypothesis to provide a universal explanation for the TSR^{21,24,31,34,35,37,39,40}”

Line 113: consider replacing “theories” to “hypotheses”

ANSWER: Done as requested.

Line 126-127: Just to clarify the fish in the AG are currently smaller than the GO fish, or projected to get smaller? If the former then I understand the relevance to compare the two population but since its written “estimated to be 14-20% smaller” it wasn't completely clear.

ANSWER: Sorry for the lack of clarity here. The AG fishes are currently 14-40% smaller than GO fishes (Brandl et al 2020, D'Agostino et al 2021). We have now rephrased this sentence to read: "*Importantly, fishes within the AG are also 14-40% smaller at maximum size compared to populations of the same species within the GO (including cryptic and larger bodied fishes not targeted by fishers^{66,72}) highlighting that the projected 'shrinking of fishes' phenomenon²⁰ is already in full effect within this ecosystem*".

Line 139-140: Is locomotor performance predicted to scale in a similar way to metabolism?

ANSWER: Excellent question. Given the tight connection between swimming performance (as the most energetically costly activity fish perform) and aerobic surplus (i.e. aerobic scope), locomotor performance is expected to follow metabolic responses across temperatures, as has previously been shown for e.g. coral reef fishes across cold to warm conditions. This information has now been added to line 107.

Figure 1: This map shows maximum sea surface temperatures but do you have information about the duration of these maximum temperatures or the variability of temperatures at these two locations, i.e. how often were the fish experiencing these temperatures, if the fish rarely experience these maximum temperatures or for only short durations will they have adaptations to them? Some information/discussion around this would be nice to include.

ANSWER: The maximum summer temperatures are typically found between late May until September and may persist for months. As such, the duration of these maximum temperatures is long enough that fish cannot survive by ceasing activity or feeding, and therefore must be able to maintain some level of performance. This information is now added to line 156: "*These summer extremes usually persist within 2.0°C of maximum from late May until September^{73,78,79}*".

Lines 166-171: what is the size at maturity for these two species and the maximum sizes? Does the size range you used represent this full range and if not why was this range chosen? Also if the fish from the AG are smaller than by size matching fish then are you excluding the biggest fish in the GO population as the assumption would be they don't exist in the AG?

ANSWER: This is a good question and presently unknown for most species in the AG. Since elevated temperatures are thought to simultaneously cause earlier maturation and smaller maximum sizes, we can make some tentative extrapolations to try to answer this question. Outside the AG, *Scolopsis ghanam* is known to start maturing at ~8cm SL with 50% fully mature at ~12cm SL, and reach a maximum size of 15cm SL (see e.g. Nzioka et al 1983; Russel 1990). Inside the AG, the size of maturity of *S. ghanam* is unknown, but survey observations suggest a 10-40% size reductions relative to GO (Vaughan et al 2021; pers. obs.). We do have a bit more information about *Lutjanus Ehrenbergii*: This species is thought to mature prior to age two, which occurs at ~8cm SL in GO (based on D'Agostino et al 2021; Ketabi & Jamili 2017). This species reaches a maximum size of ~22cm SL in GO, but only ~17cm SL inside the AG.

As the primary topic of interest for this study was fishes within the AG, the size range used in this study represent the full range of mature individuals available to catch from the AG. When testing cardio-respiratory functions in fish it is critically important to match fish sizes across treatments and with the volume of respirometers in order to maximize accuracy of oxygen consumption measures while avoiding confounding factors of using disparate equipment across treatments (see Svendsen et al 2015; Steffensen 1989; and Clark et al 2013 for lengthy discussion of the topic). As such, best-practices made it necessary to size match our collections across AG and GO to the greatest extent possible to facilitate direct comparisons. We therefore collected the widest possible adult-size ranges in AG, which did exclude the

largest possible fish sizes outside the AG as these cannot be size matched (see e.g. D'Agostino et al 2021).

Lines 175-182: How many fish were housed in each tank?

How many individuals were there within each temperature treatment and were they of equal sizes within treatments?

ANSWER: We apologize for the lack of clarity. Line 185 now reads: “Fish were housed in groups of four fish per tank and tagged with visible elastomer tags (Northwest Marine Technology, Inc.) to allow individual identification”. Additionally, the number of individuals in each treatment has been added to the statistical Tables S1 and S2, stating “A total of 44 *L. ehrenbergii* were used for final analyses, equating to 11, 16, and 16 individuals at 27.0, 31.5 and 35.5°C respectively” and “A total of 40 *S. ghanam* were used for final analyses, equating to 17, 17, and 6 individuals at 27.0, 31.5 and 35.5°C respectively”. Line 286 now states: “In accordance with climate change projections for shrinking body size of fishes in warmer waters^{17–20,24}, adult AG fishes are consistently smaller than GO fishes of the same age^{71,77}. Despite significant effort to collect a similar range of sizes from both the AG and the GO region, collected AG fishes were generally smaller in mass (*L. ehrenbergii* 37.6±4.3g versus 62.8±3.4g; *S. ghanam* 34.3±1.8g versus 48.4±4.8g) and standard length (SL, *L. ehrenbergii* 11.5±0.3cm versus 13.6±0.3cm; *S. ghanam* 10.9±0.2cm versus 12.1±0.4cm, mean ± S.E.M.), but were equal within treatments for each region (Table S3)”.

Line 183: Since fish were collected at different time periods then I assume the experiments were also done at different times, it would be useful to include a timeline (in the supp) to show this. And do you think seasonality would have affected the performance of fish (not only in terms of temperature).

ANSWER: To avoid holding fishes for months on end after collections, all experiments were staggered throughout the year in concert with collections. We have highlighted the timeline in line 180, stating that “These temperatures corresponded to the approximate annual mean and summer max temperatures found in AG (27.0 and 35.5°C) and GO (27.0 and 31.5°C) regions⁷¹, and allowed collections and corresponding trials to occur during the shoulder seasons and summer of 2018 and 2019”. Incidentally, the only seasonal impact of performance could have been reproductive status. However, none of our fishes were found to be heavily gravid, likely due to the fact that both species appear to spawn during cooler periods within the AG, with e.g. *S. ghanam* reported to spawn during winter (see Nzioka et al 1983) while *L. ehrenbergii* predominantly spawns during the early spring (Grandcourt et al. 2011; <https://fishbase.mnhn.fr/summary/Lutjanus-ehrenbergii.html>).

Lines 184-191: I understand that it wasn't possible to collect fish from the GO at 35.5 but would it not have been more comparable to have done the same acclimation on the fish from AG (i.e. collected extra fish at 31.5 and increased temp to 35.5 and held for 3 weeks)?

ANSWER: This is an interesting idea, and one we also considered. Given the main premise of our study was to understand consequences of survival at the extreme temperatures of the AG, we opted to collect our fishes at the natural peak of 35.5C to keep everything as natural as possible for all aspects of this study within the AG. Under ideal conditions we would have liked to compare performance of “wild caught 35.5C AG fish” to “AG fishes collected at 31.5 and then acclimated to 35.5C in the lab”. However, this was impossible for permitting and logistical reasons of working within this region, as we were limited in the number of fish we could collect for each temperature examined. We do want to acknowledge this putative limitation, and line 594 therefore reads: “...although the fact that only the AG individuals could

be collected and directly tested at 35.5°C renders some uncertainty for the mass scaling exponent at this temperature”.

Lines 292-296: Would be nice to include in the supp a figure of how SMR and MMR were determined. Do you think SMR is reached in the timeframe used in these experiments? 8 hours from being placed in a respirometry chamber and, during daylight hours I don't imagine will give a true SMR. Do you have any supplementary data to support this? Otherwise I would suggest calling it RMR. Some extra details on how the MMR was calculated would also be good to include, e.g. was the whole slope for the entire Ucrit speed used? Or a shorter time interval? Or something else..

ANSWER: Thank you. We have now included a supplemental methods figure depicting the determination of SMR and MMR. Importantly, numerous studies of tropical coral reef fishes have repeatedly shown that tropical fishes typically reach true SMR within 4-6 hrs (see e.g. Roche et al 2013, Binning et al 2015, Johansen & Jones 2011, Rummer et al 2014), whereas cold-water species often require much longer (often 24-48hrs). For all our studies, oxygen consumption was continuously recorded and no experiment was started until MO₂ had plateaued and stabilized at a steady-state level, which always occurred within ~3-5hrs. In addition to our swimming respirometry using 8+ hrs acclimation, we have an upcoming separate manuscript in which we exposed both of these species (and several other species) to resting respirometry protocols, in which the individuals were left to recover in the resting respirometers overnight for ~18hrs. These data also confirm that SMR is typically reached within 3-5hrs in our species, as is typical at high temperatures. As such, our acclimation period was almost twice that needed to reach SMR. We have therefore added multiple supporting references to bolster this sentence in the manuscript, including a reference to our new supplemental methods Figure SM1. It is important to recognize, however, that swimming respirometry never records SMR directly. The lowest oxygen consumption that can be measured in swimming respirometry is MO₂ at the lowest swimming speed recorded (here a fish swimming at 0.5 body length per second, which is often referred to as e.g. MO₂@0.5Bl s⁻¹). Contrary to resting respirometry where SMR is measured directly, in swimming respirometry SMR is extrapolated from a regression curve of swimming speed over MO₂, with a swimming speed of zero equating to SMR (i.e. oxygen consumption of a fish at rest). This follows best-practices for the field and provides accurate and repeatable estimates of SMR that are equivalent to resting respirometry protocols (see Roche et al 2013, Rummer et al 2016, Svendsen et al 2016, Steffensen et al 1984, Steffensen 1989). Similarly, MMR was extrapolated following best-practices by using the highest oxygen uptake recorded across a full measuring period. We apologize for not being clear on this point in the original manuscript. We have now added extra detail in line 312, which reads: “..Standard metabolic rate (i.e. SMR, oxygen uptake at rest) was then extrapolated from a nonlinear regression of oxygen uptake measures following $y = a + be^{cx}$ (with SMR as the y-axis intercept¹⁰⁴). MMR was taken as the highest recorded oxygen uptake across a single measuring period, which always occurred either at or immediately before U_{crit}. Aerobic scope was calculated as $AS = MMR - SMR$ ”.

Lines 308: “linear mixed effect model”

ANSWER: Thank you for this correction. We have updated “linear effect mixed model” to “linear mixed effect model” throughout the manuscript.

Fig.2: long term tolerance limit less than 35.5 for S.ghanam – suggests there is some acclimation/adaptive change between populations. What is the thermal tolerance (short or long term) for these two species and has it previously been shown to differ between these two populations?

ANSWER: Thank you, this is exactly what we conclude as well and a specific topic of our discussion. See line 528: “Contrary to *L. ehrenbergii*, *S. ghanam* showed clear signs of a population level response to survival at the elevated temperatures of the AG both in terms of oxygen supply capacity and swimming performance. While this species also showed increasing metabolic demand for swimming with rising temperature, only the AG individuals were able to maintain AS up to 35.5°C. This metabolic surplus allowed the AG individuals to fuel critical swimming activities at 35.5°C, whereas the GO individuals demonstrated a complete inability to survive at the same temperature. Interestingly, the GO individuals also suffered >20.0% reductions in AS and concurrent 17-22% reductions in all swimming performance metrics at 31.5°C, suggesting that the ecology of this species allow individuals to survive prolonged periods of reduced swimming performance during summer. Indeed, previous work has shown that some fishes in the AG region broaden their diet in the spring but then narrow their diets in the summer¹²⁴, while others upregulate foraging and energy storage in the spring, but then downregulate costly swimming activities during summer¹²⁵. This latter trait was mirrored in the AG *S. ghanam* as there were no differences in metabolic and swimming performance of the GO individuals at 31.5°C and the AG individuals at 35.5°C, suggesting that population level responses to elevated temperatures have led to a significant right-shift in the thermal performance window of this species¹²¹, but no other changes in the ecological performance or survival strategy during peak summer”. To our knowledge, this is the first study to examine thermal tolerance/performance limits in these species.

Results: for me all the stats in the text of the results make it hard to read and understand, although perhaps unconventional maybe these can be move to a table/supp to make it clearer?

ANSWER: We concur that our results are complex and have therefore placed all statistical values in Tables S1-3 and are instead presenting results in plain language, indicating main points, hypotheses and main findings. We believe the new presentation is clearer and easier to comprehend. However, as this approach does not convey detailed results with exact accompanying statistics as many readers are accustomed, we have opted to also provide an updated version of the detailed written results section in our supplemental documentation. This way, the reader can now choose between a plain language results section in the main manuscript, or an accompanying in-depth results section in the supplemental material. We hope this solution is satisfactory.

Figures: would be useful to include sample size in figure legend and for the box plots to have the data points overlaid so its easier to see the distribution of the data. Figure 1 and 2 could also have a description of the dashed grey lines and maybe they aren't needed between the SMR and MMR, unless there for a specific purpose?

ANSWER: Thank you for these suggestions. To avoid repetition, we have included sample sizes in the statistical output Tables S1 and S2. Additionally, we have now overlaid the raw data points in the box plots as suggested thereby allowing all raw data to be seen, included a description of the dashed grey lines, and removed these between SMR and MMR. We believe the figures are improved as a result.

Figure 4,5 and 6: I understand the reasoning for pooling the data from the two sites but if you looked at each site independently do the relationships remain the same? For *S.ghanam* in particular since you miss data from the highest temp at one of the sites you increase the variance of the other two temperatures compared to this one, does your statistical test account for this? Also you are aiming to look at the allometry between traits but in two very different populations, one that has the “full” size range and has not undergone intense selection for high temperatures and one with a overall smaller size which has already undergone selection, would

you expect these two to scale equally? Some justification or clarification of the reasoning for this would be nice to include.

ANSWER: Yes, we explicitly tested for this and the relationships remained the same. Importantly, we only pooled data when scaling relationships were the same for each site independently, thereby increasing statistical strength for general conclusions. We also explicitly examined model fit for all analyses, only using the most parsimonious models. We do acknowledge that this was not abundantly clear in our old representation of results, but have now followed the suggestions of both reviewers to move all statistical outputs to separate tables and only present general patterns in the text of the results section. These statistical comparisons and validations are presented for view. We hope our findings are now easier to digest. Ps, for those preferring a more traditional presentation of results (i.e., text mixed with statistical data) we have included a fully detailed results section in the supplemental information.

Line 486: “two adaptive pathways”, even though this is an introduction to the results you could state what these two pathways are here and then go into more detail afterwards.

ANSWER: Thank you for this suggestion. We have now included an introductory statement about these pathways, before going into details afterwards. The sentence now reads (line 470): “*Using the worlds warmest coral reefs in the Arabian Gulf (AG) as a natural laboratory for ocean warming, our data reveals two adaptive pathways across 10 metabolic and swimming performance metrics that appear to have facilitated survival at elevated temperatures, including a lower-than-expected rise in basal metabolic demands and a right-shifted thermal window facilitating a capacity to maintain aerobic scope and critical swimming speeds into higher temperatures*”.

Line 493: you mention a direct connection between these two locations, is there evidence of genetic mixing between your populations or are they genetically distinct?

ANSWER: This is an extremely interesting question and one which has yet to be answered. A recent study of *Lutjanus ehrenbergii* used tissues from fish markets to suggested limited genetic differentiation within the Arabian Peninsula (i.e. between nearby AG and GO sites) but clear differentiation to the Red Sea and Gulf of Aden (Griffiths et al 2024). However, most fish markets in AG and GO share market products, rendering some uncertainty about the origin of tissues examined (Griffiths et al 2024). We do not have genetic analyses of *Scolopsis ghanam* examined here, but another, as of yet unpublished, analysis of a similar species, *Cheilodipterus novemstriatus*, do show genetic separation between AG and GO. As such, we are unable to answer this question although it is reasonable to theorize that other species in the AG will also show genetic separation. To refrain from erroneously categorizing these as genetically distinct populations, even if likely, we refer to “regional differences” and “population-level” responses throughout the manuscript as opposed to distinct “population-specific” responses.

Line 494: The fact that 12% of species do exist in the AG suggests these species have been able to adapt or acclimate to the warmer temperatures, even if the others haven’t (despite your prediction in the introduction). Some discussion around this would be beneficial.

ANSWER: Thank you for this suggestion. In addition to several paragraphs discussing the physiological underpinnings for this interesting pattern (line 485 onward) we have now included a discussion of some ecological underpinnings for survival in the AG related to habitat and dietary plasticity. A new section in line 637 reads: “*Interestingly, the less than 12% subset of GO species that have managed to survive in AG mirror these ecological commonalities: They are typically species with high habitat and dietary plasticity⁷³, and able to alter foraging strategies and foraging rates in response to intrinsic and/or extrinsic*

factors. Indeed, altered foraging patterns of AG fishes have been documented both in-situ and in-vivo, with e.g., *Pomacanthus maculosus* (a typical herbivore / omnivore^{124,152,153}) feeding predominantly on coral and sponges in AG¹²⁴, while *Pomacentrus trichourus* (a typical planktivore¹⁵⁴) shift to benthic resources during the AG summer^{124,155}. Importantly, these patterns have developed across numerous generations within the >6,000-year history of the AG, highlighting that...”.

Lines 510-520: The conclusion here that *L. ehrenbergii* doesn't have any adaptive pathways for surviving high temperatures can not be said as the authors clearly show that this fish is still within its optimal temperature range at the highest temperatures tested. If the species was tested at a higher temperature still then maybe differences could be seen between populations (like with the other species tested).

ANSWER: We acknowledge that this statement is only true within the confined temperatures examined here. We apologize for the lack of clarity and have now added new text (underlined below) to highlight that we are only referring to the temperature range found within AG. Line 516 now reads: “Explicitly, *L. ehrenbergii* showed no population level signs of adaptation to life at the elevated temperatures found within the AG, instead appearing to thrive in the AG due to a high intrinsic species-specific tolerance to temperatures reaching 35.5°C”

References:

- Binning, S. A., Ros, A. F., Nusbaumer, D., & Roche, D. G. (2015). Physiological plasticity to water flow habitat in the damselfish, *Acanthochromis polyacanthus*: linking phenotype to performance. *PLoS One*, *10*(3), e0121983.
- Brown, J.H., Gillooly, J.F., Allen, A.P., Savage, V.M. & West, G.B. (2004). Toward a metabolic theory of ecology. *Ecology*, *85*, 1771–1789.
- Clark, T. D., Sandblom, E., & Jutfelt, F. (2013) Aerobic scope measurements of fishes in an era of climate change: respirometry, relevance and recommendations. *Journal of Experimental Biology*, *216*(15), 2771-2782. Doi:10.1242/jeb.084251
- Clarke, A., & Johnston, N. M. (1999). Scaling of metabolic rate with body mass and temperature in teleost fish. *Journal of animal ecology*, *68*(5), 893-905.
- D’Agostino, D., Burt, J. A., Santinelli, V., Vaughan, G. O., Fowler, A. M., Reader, T., ... & Feary, D. A. (2021). Growth impacts in a changing ocean: insights from two coral reef fishes in an extreme environment. *Coral Reefs*, *40*(2), 433-446. Doi: 10.1007/s00338-021-02061-6
- Grandcourt, E., Al Abdessalaam, T. Z., Francis, F., & Al Shamsi, A. (2011) Demographic parameters and status assessments of *Lutjanus ehrenbergii*, *Lethrinus lentjan*, *Plectorhinchus sordidus* and *Rhabdosargus sarba* in the southern Arabian Gulf. *Journal of Applied Ichthyology*, *27*(5), 1203-1211.
- Griffiths, MH, Christopher M Wade, Daniele D’Agostino, Michael L Berumen, John A Burt, Joseph D DiBattista, David A Feary, Phylogeography of a commercially important reef fish, *Lutjanus ehrenbergii*, from the coastal waters of the Arabian Peninsula, *Biological Journal of the Linnean Society*, 2024;, blad170, <https://doi.org/10.1093/biolinnean/blad170>
- Hernández, Adrián V., Ewout W. Steyerberg, J. Dik F. Habbema (2004) Covariate adjustment in randomized controlled trials with dichotomous outcomes increases statistical power and reduces sample size requirements. *Journal of clinical epidemiology*, *57.5*, 454-460.

- Johansen, J. L., & Jones, G. P. (2011). Increasing ocean temperature reduces the metabolic performance and swimming ability of coral reef damselfishes. *Global Change Biology*, 17(9), 2971-2979.
- Johansen, J. L., Nadler, L. E., Habary, A., Bowden, A. J., & Rummer, J. (2021). Thermal acclimation of tropical coral reef fishes to global heat waves. *Elife*, 10, e59162. <https://doi.org/10.7554/eLife.59162>
- Kahan, B.C., Jairath, V., Doré, C.J. *et al.* (2014) The risks and rewards of covariate adjustment in randomized trials: an assessment of 12 outcomes from 8 studies. *Trials*, 15, 139. <https://doi.org/10.1186/1745-6215-15-139>
- Ketabi, R., & Jamili, S. (2017) *Lutjanus ehrenbergii* (Peters, 1869). <http://hdl.handle.net/1834/36055>
- Killen, S. S., Atkinson, D., & Glazier, D. S. (2010). The intraspecific scaling of metabolic rate with body mass in fishes depends on lifestyle and temperature. *Ecology letters*, 13(2), 184-193.
- Momal R, Li H, Trichelair P, Blum MGB, Balazard F. (2023) More efficient and inclusive time-to-event trials with covariate adjustment: a simulation study. *Trials*, 24(1), 380. doi: 10.1186/s13063-023-07375-0. PMID: 37280655; PMCID: PMC10245605.
- Nzioka, R. M. (1983). Some biological aspects of (*Scolopsis ghanam* [Forsk.] perciformes: Scolopsidae). *East African Agricultural and Forestry Journal*, 48(1-4). Doi: 10.1080/00128325.1982.11663104
- Queiros, Q., McKenzie, D. J., Dutto, G., Killen, S., Saraux, C., & Schull, Q. (2024). Fish shrinking, energy balance and climate change. *Science of the Total Environment*, 906, 167310.
- Reist, J. D. (1985). An empirical evaluation of several univariate methods that adjust for size variation in morphometric data. *Canadian Journal of Zoology*, 63(6), 1429-1439
- Roche, D. G., Binning, S. A., Bosiger, Y., Johansen, J. L., & Rummer, J. L. (2013). Finding the best estimates of metabolic rates in a coral reef fish. *Journal of Experimental Biology*, 216(11), 2103-2110.
- Rummer, J. L., Binning, S. A., Roche, D. G., & Johansen, J. L. (2016). Methods matter: considering locomotory mode and respirometry technique when estimating metabolic rates of fishes. *Conservation Physiology*, 4(1), cow008.
- Rummer, J. L., Couturier, C. S., Stecyk, J. A., Gardiner, N. M., Kinch, J. P., Nilsson, G. E., & Munday, P. L. (2014). Life on the edge: thermal optima for aerobic scope of equatorial reef fishes are close to current day temperatures. *Global change biology*, 20(4), 1055-1066.
- Russell, B.C. (1990) FAO species catalogue. Vol. 12. Nemipterid Fishes of the World. (Threadfin breams, Whiptail breams, Monocle breams, Dwarf monocle breams, and Coral breams). Family Nemipteridae. An Annotated and Illustrated Catalogue of Nemipterid Species known to Date. FAO Fisheries Synopsis. No. 125, Volume 12. Rome, FAO. 149 p., VIII plates. <https://www.fao.org/3/t0416e/t0416e.pdf>
- Skeeles, M. R., & Clark, T. D. (2023). Evidence for energy reallocation, not oxygen limitation, driving the deceleration in growth of adult fish. *Journal of Experimental Biology*, 226(13).
- Skeeles, M. R., Scheuffele, H., & Clark, T. D. (2023) Supplemental oxygen does not improve growth but can enhance reproductive capacity of fish. *Proceedings of the Royal Society B*, 290, 20231779. <https://doi.org/10.1098/rspb.2023.1779>
- Steffensen, J. F. (1989) Some errors in respirometry of aquatic breathers: how to avoid and correct for them. *Fish Physiol Biochem*, 6(1), 49-59. Doi: 10.1007/BF02995809
- Steffensen, J. F. (1989). Some errors in respirometry of aquatic breathers: how to avoid and correct for them. *Fish Physiol Biochem*, 6(1), 49-59.
- Steffensen, J. F., Johansen, K., & Bushnell, P. G. (1984). An automated swimming respirometer. *Comparative biochemistry and physiology--Part A: Physiology*, 79(3), 437-440.

Svendsen, M. B. S., Bushnell, P. G., & Steffensen, J. F. (2016) Design and setup of intermittent-flow respirometry system for aquatic organisms. *Journal of fish biology*, 88(1), 26-50. Doi: 10.1111/jfb.12797

Svendsen, M. B. S., Bushnell, P. G., & Steffensen, J. F. (2016). Design and setup of intermittent-flow respirometry system for aquatic organisms. *Journal of fish biology*, 88(1), 26-50.

Tackney, M.S., Morris, T., White, I. *et al.* (2023) A comparison of covariate adjustment approaches under model misspecification in individually randomized trials. *Trials*, 24, 14. <https://doi.org/10.1186/s13063-022-06967-6>

Tsiatis, Anastasios A., et al. (2008) Covariate adjustment for two-sample treatment comparisons in randomized clinical trials: a principled yet flexible approach. *Statistics in medicine*, 27.23, 4658-4677.

White, C. R., Phillips, N. F., & Seymour, R. S. (2006). The scaling and temperature dependence of vertebrate metabolism. *Biology letters*, 2(1), 125-127.

Reviewers' Comments:

Reviewer #1:

Remarks to the Author:

Thank you for addressing my suggestions and providing such a detailed response. I am satisfied that my comments and suggestions have been addressed and recommend this manuscript for publication. While I do not necessarily fully agree with the strength of evidence or support for the PASLED hypothesis, I do agree with the authors that it is good to get these ideas out and for further discussion. Well done on this excellent study.

Reviewer #2:

Remarks to the Author:

This is now my second time reviewing this MS and firstly I commend the authors for the thorough responses and satisfying answers to previous questions and I think the changes made to the MS have significantly increased its clarity and now reads very well. I only have a couple of minor comments.

I think the main one is still whether or not the new hypothesis should be included in this paper. Despite the justifications given I agree with the other reviewer that it detracts from the very interesting results of this paper and do think it would be better as a stand-alone perspective.

The right-shifted thermal performance window could be shown in a conceptual figure in the discussion to make this clearer. In fact, a summary conceptual figure with all the main results could be useful to help the reader.

Figures overall look much better, as does the description of the results but a couple of tiny error in Figure 5. Top left figure – “red” line appears pink. Bottom left figure – remove NS next to points so its consistent with the other figures.

First sentence of the abstract appears almost identical to first sentence of discussion. Recommend rewording one of them.

RESPONSE TO REVIEWERS' COMMENTS

Reviewer #1 (Remarks to the Author):

Thank you for addressing my suggestions and providing such a detailed response. I am satisfied that my comments and suggestions have been addressed and recommend this manuscript for publication. While I do not necessarily fully agree with the strength of evidence or support for the PASLED hypothesis, I do agree with the authors that it is good to get these ideas out and for further discussion. Well done on this excellent study.

ANSWER: Thank you very much for the helpful comments and reviews. We believe the manuscript is greatly improved as a result!

Reviewer #2 (Remarks to the Author):

This is now my second time reviewing this MS and firstly I commend the authors for the thorough responses and satisfying answers to previous questions and I think the changes made to the MS have significantly increased its clarity and now reads very well. I only have a couple of minor comments.

ANSWER: Thank you. We fully agree and greatly appreciate the suggestions provided!

I think the main one is still whether or not the new hypothesis should be included in this paper. Despite the justifications given I agree with the other reviewer that it detracts from the very interesting results of this paper and do think it would be better as a stand-alone perspective.

ANSWER: Thank you. While we appreciate this notion, as stated previously we firmly believe this is the right venue for our theory.

The right-shifted thermal performance window could be shown in a conceptual figure in the discussion to make this clearer. In fact, a summary conceptual figure with all the main results could be useful to help the reader.

ANSWER: Thank you. We contemplated this idea extensively, but given the multitude of data results across 10 different performance measures x two species x two regions, we believe a single conceptual figure would become too complex or too abstract to effectively help the reader.

Figures overall look much better, as does the description of the results but a couple of tiny error in Figure 5. Top left figure – “red” line appears pink. Bottom left figure – remove NS next to points so its consistent with the other figures.

ANSWER: Thank you for catching these errors. They have now been corrected.

First sentence of the abstract appears almost identical to first sentence of discussion. Recommend rewording one of them.

ANSWER: Thank you for bringing this to our attention. The first line of the Discussion has now been rephrased to read “The consequences of ocean warming for fish and fisheries is a topic of vigorous debate”